# Training Spiking Neural Networks
# with Local Tandem Learning

**Qu Yang**[1], **Jibin Wu**[2],[*] **Malu Zhang**[3]**, Yansong Chua**[4]**, Xinchao Wang**[1]**, Haizhou Li**[5,6,1]
[1]National University of Singapore
[2]The Hong Kong Polytechnic University
[3]University of Electronic Science and Technology of China
[4]China Nanhu Academy of Electronics and Information Technology
[5]The Chinese University of Hong Kong, Shenzhen, China
[6]Kriston AI, Xiamen, China

## Abstract

Spiking neural networks (SNNs) are shown to be more biologically plausible and energy efficient over their predecessors. However, there is a lack of an efficient and generalized training method for deep SNNs, especially for deployment on analog computing substrates. In this paper, we put forward a generalized learning rule, termed Local Tandem Learning (LTL). The LTL rule follows the teacher-student learning approach by mimicking the intermediate feature representations of a pre-trained ANN. By decoupling the learning of network layers and leveraging highly informative supervisor signals, we demonstrate rapid network convergence within five training epochs on the CIFAR-10 dataset while having low computational complexity. Our experimental results have also shown that the SNNs thus trained can achieve comparable accuracies to their teacher ANNs on CIFAR-10, CIFAR-100, and Tiny ImageNet datasets. Moreover, the proposed LTL rule is hardware friendly. It can be easily implemented on-chip to perform fast parameter calibration and provide robustness against the notorious device non-ideality issues. It, therefore, opens up a myriad of opportunities for training and deployment of SNN on ultra-low-power mixed-signal neuromorphic computing chips.

## 1 Introduction

Over the last decade, artificial neural networks (ANNs) have improved the perceptual and cognitive capabilities of machines by leaps and bounds, and have become the de-facto standard for many pattern recognition tasks including computer vision [30, 52, 53, 62, 35], speech processing [60, 39], language understanding [3], and robotics [51]. Despite their superior performance, ANNs are computationally expensive to be deployed on ubiquitous mobile and edge computing devices due to high memory and computation requirements.

Spiking neural networks (SNNs), the third-generation artificial neural networks, have gained growing research attention due to their greater biological plausibility and potential to realize ultra-low-power computation as observed in biological neural networks. Leveraging the sparse, spike-driven computation and fine-grain parallelism, the fully digital neuromorphic computing (NC) chips like TrueNorth [2], Loihi [11], and Tianjic [42], that support the efficient inference of SNNs, have indeed demonstrated orders of magnitude improved power efficiency over GPU-based AI solutions. Moreover, the emerging in situ mixed-signal NC chips [54, 47], enabled by nascent non-volatile technologies, can further boost the hardware efficiency by a large margin over the aforementioned digital chips.

---

[*]Corresponding Author: jibin.wu@polyu.edu.hk

36th Conference on Neural Information Processing Systems (NeurIPS 2022).

Despite remarkable progress in neuromorphic hardware development, how to efficiently and effectively train the core computational model, spiking neural network, remains a challenging research topic. It, therefore, impedes the development of efficient neuromorphic training chips as well as the wide adoption of neuromorphic solutions in mainstream AI applications. The existing training algorithms for deep SNNs can be grouped into two categories: ANN-to-SNN conversion and gradient-based direct training.

For ANN-to-SNN conversion methods, they propose to reuse network weights from more easily trainable ANNs. This can be viewed as a specific example of Teacher-Student (T-S) learning that transfers the knowledge from a teacher ANN to a student SNN in the form of network weights. By properly determining the neuronal firing threshold and initial membrane potentials for SNNs, recent studies show that the activation values of ANNs can be well approximated with the firing rate of spiking neurons, achieving near-lossless network conversion on a number of challenging AI benchmarks [7, 49, 14, 22, 21, 46, 34, 12, 4, 5, 63]. Nevertheless, these network conversion methods are developed solely based on the non-leaky integrate-and-fire (IF) neuron model and typically require a large time window so as to reach a reliable firing rate approximation. It is, therefore, not straightforward and efficient to deploy these converted SNNs onto the existing neuromorphic chips.

In another vein of research, the gradient-based direct training methods explicitly model each spiking neuron as a self-recurrent neural network and leverage the canonical Backpropagation Through Time (BPTT) algorithm to optimize the network parameters. The non-differentiable spiking activation function is typically circumvented with continuous surrogate gradient (SG) functions during error backpropagation [50, 58, 59, 38, 8, 15, 45, 61]. Despite their compatibility with event-based inputs and different spiking neuron models, they are computational and memory inefficient to operate in practice. Moreover, the gradient approximation error introduced by these SG functions tends to accumulate over layers, causing significant performance degradation in the face of the deep network structure and short time window [57].

In general, SNN learning algorithms can be categorized into off-chip learning [16, 64] and on-chip learning [9, 41, 43]. Almost all of the direct SNN training methods discussed above belong to the off-chip learning category. Due to the lack of effective ways to exploit the high level of sparsity in spiking activities and the requirement to store non-local information for credit assignment, these off-chip methods exhibit very low training efficiency. Moreover, due to notorious device non-ideality problems [6], the actual network dynamics will deviate from the off-chip simulated ones, causing the accuracy of off-chip trained SNNs degrades significantly when deployed onto the analog computing substrates [44, 24, 37, 1]. To address these problems, recent work proposes on-chip learning algorithms in the form of local Hebbian learning [11, 41, 28] and approximation of gradient-based learning [32, 40, 10, 19], while the effectiveness of these algorithms had only been demonstrated on simple benchmarks, such as MNIST and N-MNIST datasets.

To address the aforementioned problems in SNNs training and hardware deployment, we put forward a generalized SNN learning rule in this paper, which we referred to as the Local Tandem Learning (LTL) rule. The LTL rule takes the best of both ANN-to-SNN conversion and gradient-based training methods. On one hand, it makes good use of highly effective intermediate feature representations of ANNs to supervise the training of SNNs. By doing so, we show that it can achieve rapid network convergence within five training epochs on the CIFAR-10 dataset with low computational complexity. On the other hand, the LTL rule adopts the gradient-based approach to perform knowledge transfer, which can support different neuron models and achieve rapid pattern recognition. By propagating gradient information locally within a layer, it can also alleviate the compounding gradient approximation errors of the SG method and lead to near-lossless knowledge transfer on CIFAR-10, CIFAR-100, and Tiny ImageNet datasets. Moreover, the LTL rule is designed to be hardware friendly, which can perform efficient on-chip learning using only local information. Under this on-chip setting, we demonstrate that the LTL rule is capable of addressing the notorious device non-ideality issues of analog computing substrates, including device mismatch, quantization noise, thermal noise, and neuron silencing.

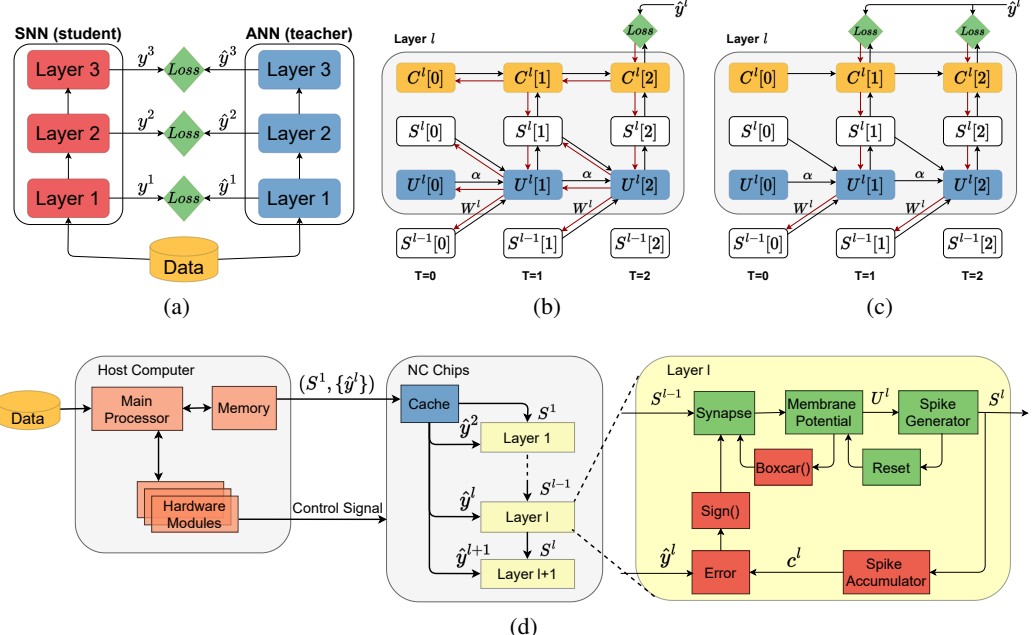

Figure 1: Illustration of the proposed LTL rule and its on-chip implementation. (a) The LTL rule follows the teacher-student learning approach, whereby the SNN tries to mimic the feature representation of a pre-trained ANN through local loss functions. (b) Computational graph of the offline LTL rule. (c) Computational graph of the online LTL rule. (d) Functional block diagram of the proposed on-chip implementation, where the host computer transfers the control signal and training data (input spike train, layerwise targets) to NC chips. The proposed SNN on-chip learning circuit consists of two parts: spiking neuron (green) and learning circuits (red).

## 2 Methods

### 2.1 Spiking Neuron Model

To demonstrate the proposed LTL rule is compatible with different spiking neuron models, we base our study on both non-leaky integrate-and-fire (IF) [48] and leaky integrate-and-fire (LIF) neuron models [18], whose neuronal dynamics can be described by the following discrete-time formulation:

$$U_i^l[t] = \alpha U_i^l[t-1] + I_i^l[t] - \vartheta S_i^l[t-1] \tag{1}$$

with

$$I_i^l[t] = \sum_j w_{ij}^{l-1} S_j^{l-1}[t-1] + b_i^l \tag{2}$$

where $U_i^l[t]$ and $I_i^l[t]$ refer to the subthreshold membrane potential of and input current to neuron $i$ at layer $l$, respectively. $\alpha \equiv exp(-dt/\tau_m)$ is the membrane potential decaying constant, wherein $\tau_m$ is the membrane time constant and $dt$ is the simulation time step. For IF neuron, $\alpha$ takes a value of 1. $\vartheta$ denotes the neuronal firing threshold. $w_{ij}^{l-1}$ represents the connection weight from neuron $j$ of the preceding layer $l-1$ and $b_i^l$ denotes the constant injecting current to neuron $i$. $S_i^l[t-1]$ indicates the occurrence of an output spike from neuron $i$ at time step $t-1$, which is determined according to the spiking activation function as per

$$S_i^l[t] = \Theta\left(U_i^l[t] - \vartheta\right) \quad \text{with} \quad \Theta(x) = \begin{cases} 1, & \text{if} \quad x \geq 0 \\ 0, & \text{otherwise} \end{cases} \tag{3}$$

## 2.2 Local Tandem Learning

As illustrated in Figure 1(a), the LTL rule follows the T-S learning approach [20], whereby the intermediate feature representation of a pre-trained ANN is transferred to SNN through layer-wise local loss functions. In contrast to the ANN-to-SNN conversion methods, we establish the feature representation equivalence at the neuron level rather than at the synapse level, which provides the flexibility for choosing any neuron model to be used in the SNN. On the other hand, with the proposed spatially local loss function, we simplify the spatial-temporal credit assignment required in the end-to-end direct training methods, which can dramatically improve the network convergence speed and meanwhile reduce the computational complexity. In the following, we introduce two versions of the LTL rule, offline and online, depending on whether the temporal locality constraint is imposed.

**Offline Learning** Following the T-S learning approach, we consider the intermediate feature representation of a pre-trained ANN as the knowledge and train an SNN to reproduce an equivalent feature representation via layer-wise loss functions. In particular, we establish an equivalence between the normalized activation values of an ANN and the global average firing rates of an SNN. To reduce the discrepancy between these two quantities, we adopt the mean square error (MSE) loss function and apply it separately for each layer. Thus, for any layer $l$, the local loss function $\mathcal{L}^l$ is defined as follows

$$\mathcal{L}^l\left(\hat{y}^l, y^l[T_w]\right) = \left|\left|\frac{\hat{y}^l}{y_{norm}} - \frac{C^l[T_w]}{T_w}\right|\right|_2^2 \tag{4}$$

where $\hat{y}^l$ is the output of ANN layer $l$. $y_{norm}$ is a normalization constant that takes the value of $99th$ or $99.9th$ percentile across all $\hat{y}_i^l$. This can alleviate the effect of outliers compared to using the maximum activation value [48]. $T_w$ is the time window size. $C^l[T_w] = \Sigma_{t=1}^{T_w} S[t]$ is total spike count.

As the computational graph shown in Figure 1(b), we adopt the BPTT algorithm to resolve the temporal credit assignment problem, and the weight gradients can be derived as

$$\frac{\partial \mathcal{L}^l}{\partial w_{ij}^l} = \sum_{t=1}^{T_w} \frac{\partial \mathcal{L}^l}{\partial U_i^l[t]} \frac{\partial U_i^l[t]}{\partial w_{ij}^l} = \sum_{t=1}^{T_w} \frac{\partial \mathcal{L}^l}{\partial U_i^l[t]} \frac{\partial U_i^l[t]}{\partial I_i^l[t]} \frac{\partial I_i^l[t]}{\partial w_{ij}^l} = \sum_{t=1}^{T_w} \frac{\partial \mathcal{L}^l}{\partial U_i^l[t]} S_j^{l-1}[t-1] \tag{5}$$

with

$$\frac{\partial \mathcal{L}^l}{\partial U_i^l[t]} = \begin{cases} \alpha \delta_i^l[t+1]\frac{\partial S_i^l[t+1]}{\partial U_i^l[t+1]} + \delta_i^l[t]\frac{\partial S_i^l[t]}{\partial U_i^l[t]} & \text{if} \quad t < T_w \\ \delta_i^l[T_w]\frac{\partial S_i^l[T_w]}{\partial U_i^l[T_w]} & \text{if} \quad t = T_w \end{cases} \tag{6}$$

where

$$\delta_i^l[t] = \frac{\partial \mathcal{L}^l}{\partial S_i^l[t]} = \begin{cases} -\vartheta \delta_i^l[t+1]\frac{\partial S_i^l[t+1]}{\partial U_i^l[t+1]} + \delta_i^l[T_w] & \text{if} \quad t < T_w \\ -\frac{2}{T_w}\left(\frac{\hat{y}_i^l}{y_{norm}} - \frac{1}{T_w}\Sigma_{t=1}^{T_w} S_i^l[t]\right) & \text{if} \quad t = T_w \end{cases} \tag{7}$$

To resolve the problem of the non-differentiable spiking activation function, we apply the surrogate gradient method, i.e., $\Theta'(x) \approx \theta'(x)$. Specifically, we adopt the boxcar function for $\theta'(x)$ that supports convenient and efficient on-chip implementation.

$$\frac{\partial S_i^l[t]}{\partial U_i^l[t]} = \theta'(U_i^l[t] - \vartheta) = \frac{1}{p} sign\left(|U_i^l[t] - \vartheta| < \frac{p}{2}\right) \tag{8}$$

where $p$ controls the permissible range of membrane potentials that allow gradients to pass through, and we tune this hyperparameter separately for each dataset. By substituting Eq. (8) into Eqs. (6) and (7), we can yield the ultimate form of weight gradients, and we can update the weights according to the stochastic gradient descent method or its adaptive variants. See Supplementary Materials Section A.1 for a more detailed derivation of the gradients to weight and bias terms.

**Online Learning** The offline LTL rule requires storing intermediate synaptic and membrane state variables so as to be used during error backpropagation, which is prohibited for on-chip learning where memory resources are limited. To address this problem, we introduce an online LTL rule, whose loss function is designed to be both spatially and temporally local. To achieve the temporal locality, we use the moving average firing rate, which can be calculated at each time step, to replace the global firing rate used in Eq. (4). It hence yields the following local loss function

$$\mathcal{L}^l[t] = \left|\left|\frac{\hat{y}^l}{y_{norm}} - \frac{C^l[t]}{t}\right|\right|_2^2 \tag{9}$$

Compared to the offline version, the gradient update is much simpler now:

$$\frac{\partial \mathcal{L}^l[t]}{\partial w_{ij}^l} = \frac{\partial \mathcal{L}^l[t]}{\partial S_i^l[t]} \frac{\partial S_i^l[t]}{\partial U_i^l[t]} \frac{\partial U_i^l[t]}{\partial w_{ij}^l} = \zeta_i^l[t] \frac{\partial S_i^l[t]}{\partial U_i^l[t]} S_j^{l-1}[t-1] \qquad (10)$$

where $\zeta^l[t]$ can be directly computed from Eq. (9):

$$\zeta_i^l[t] = \frac{\partial \mathcal{L}^l[t]}{\partial S_i^l[t]} = -\frac{2}{t} \left( \frac{\hat{y}_i^l}{y_{norm}} - \frac{1}{t} \Sigma_{k=1}^t S_i^l[k] \right) \qquad (11)$$

The computational graph of the online LTL rule is provided in Figure 1(c). It is worth noting that the first few time steps of the firing rate calculation are relatively noisy. Nevertheless, this issue can be easily addressed by treating the first few steps as the warm-up period, during which the parameter updates are not allowed (see Supplementary Materials Section C for a study on the effect of the warm-up period). By doing so, it can also reduce the overall training cost. As will be discussed in Sections 3.1 and 3.4, this online version can significantly reduce the computational complexity, while achieving a comparable test accuracy to that of the offline version.

**On-chip Implementation** To allow a convenient and efficient on-chip implementation of the proposed online LTL rule, we carefully designed the learning circuits as illustrated in Figure 1(d). The output spike count $C^l$ is updated at the spike accumulator, and it is compared to the local target $\hat{y}^l$ following the layer-wise loss function defined in Eq. (9). This error term is further feedback to the neuron to update the synaptic parameters. Note that the synaptic updates are gated by $sign(\cdot)$ and $boxcar(\cdot)$ functions, which can significantly reduce the overall number of parameter updates.

We would like to highlight that the proposed LTL learning rule is more hardware-friendly than the recently introduced hardware-in-the-loop (HIL) training approach [10, 19]. The HIL training approaches require two-way information communication, that is, (1) reading intermediate neuronal states from the NC chip to the host computer to perform off-chip training and (2) writing the updated weights from the host computer to the NC chip. Given the sequential nature of these two processes and the high implementation cost for reading neuronal states (e.g., requiring to implement costly analog-to-digital converters for analog spiking neurons), HIL training approaches are expensive to deploy in practice.

In contrast, the LTL training rule can be implemented efficiently on-chip by simultaneously extracting the layerwise targets from ANNs, running on the host computer, for data batch $i+1$ and performing on-chip SNN training for data batch $i$. This is similar to conventional ANN training, where the data preprocessing of the next data batch is performed on the CPU and meanwhile the current data batch is used for ANN training on the GPU. The only difference is that the input data is preprocessed by the pre-trained ANN to extract the targets for intermediate layers for the proposed LTL rule. Given the inference of ANN can be performed in parallel on the host computer, the overall training time is bottlenecked by the NC chip that operates in a sequential mode, where only one sample is been processed at a time. Therefore, our method has much lower hardware and time complexity.

## 3 Experiments

In this section, we evaluate the effectiveness of the proposed LTL rule on the image classification task with CIFAR-10 [29], CIFAR-100 [29], and Tiny ImageNet [55] datasets. We perform a comprehensive study to demonstrate its superiority in: 1. accurate, rapid, and efficient pattern recognition; 2. rapid network convergence with low computational complexity; 3. provide robustness against hardware-related noises. More details about the experimental datasets and implementation details are provided in the Supplementary Materials Section B, and the source code can be found at[2].

### 3.1 Accurate and Scalable Image Classification

Here, we report the classification results of LTL trained SNNs on CIFAR-10, CIFAR-100 and Tiny ImageNet datasets against other SNN learning rules, including ANN-to-SNN conversion [57, 49, 46, 34, 22, 21, 12, 4, 5, 31, 17] and direct SNN training [45, 13] methods. Given the network architectures and data preparation processes vary slightly across different work, therefore, we focus our discussions on the conversion or transfer errors between the ANNs and SNNs whenever the data is available.

---

[2]https://github.com/Aries231/Local_tandem_learning_rule

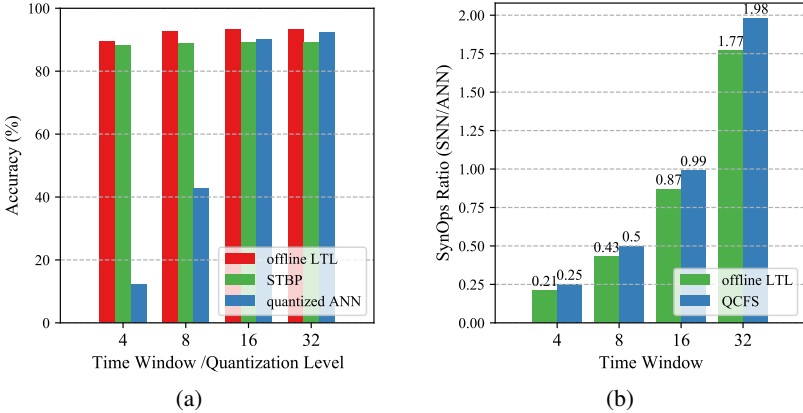

Figure 2: (a) Comparison of the network performance under different time window sizes (SNN) and quantization levels (ANN). The results are obtained on the CIFAR-10 dataset using the VGG-11 architecture. (b) The ratio of total synaptic operations between SNN and ANN as a function of the time window size.

On the CIFAR-10 dataset, as reported in Table 1, our LTL offline trained spiking VGG-11 has surpassed the accuracy of teacher ANN at $T_w = 16$, this result suggests high effectiveness of the proposed T-S learning approach in transferring the knowledge through local objectives. Moreover, the LTL rule can take advantage of a larger time window that offers a greater representation power, and further improves the performance of SNNs at $T_w = 32$. It is also worth mentioning that by transferring the knowledge in a learnable way, the proposed LTL rule is compatible with different neuron models, manifested through high accuracies achieved by both IF and LIF neuron models.

Although slightly degrades from the offline version, the online trained SNNs can still achieve comparable results to their teacher models with accuracies dropping by less than 1%, which indicates that removing temporal dependency between time steps has little impact on the training of SNNs (see Section D for a detailed study on why the online LTL rule can perform as effectively as the offline version).

For the VGG-16 model, we notice a modest accuracy drop that is more than 1% from the teacher model, which can be explained by the compounding layerwise representation errors. Fortunately, such errors can be well addressed with a larger time window. Specifically, at $T_w = 32$, the accuracy gap to the teacher ANN narrows down to only 0.13% and 0.33% respectively for LIF and IF neuron models under the offline training setting. When compared against other SOTA learning algorithms, the proposed LTL rule can achieve comparable performance in terms of both the final SNN model accuracies and the accuracy drop from ANN models. The same conclusions can also be drawn from the results of CIFAR-100 and Tiny ImageNet datasets, it hence suggests the proposed LTL rule is a generalized, scalable, and highly effective method for constructing high-performance SNNs.

### 3.2 Rapid and Efficient Pattern Recognition

To realize the goal as a power-efficient alternative to the mainstream ANNs in solving pattern recognition tasks, SNNs should make accurate and rapid predictions with a low firing rate. Here, we study whether the proposed LTL rule is able to train SNNs that can cope with a short time window. Specifically, we design experiments that progressively increase the time window $T_w$ from 4 to 32 during training, and we compare the offline LTL rule against the STBP [58] learning rule on the CIFAR-10 dataset. To ensure a fair comparison, we use the VGG-11 architecture with the same experimental settings, except that the LTL rule uses local objectives and the STBP rule uses a global objective. We note that the discrete neural representation of SNNs is equivalent to ANNs with quantized activation values [57]. Hence, we also compare our results to quantized ANNs obtained through quantization-aware training [25].

As shown in Figure 2(a), we observe that the accuracy of the quantized ANN degrades rapidly when the total quantization level is lower than 16, while both LTL and STBP trained SNNs can

Table 1: Summary of classification accuracies on CIFAR-10, CIFAR-100 and Tiny ImageNet datasets. $\Delta$ Acc. = SNN Acc. - ANN Acc.; $T_w$: time window size.

| Dataset | Method | Neuron | Architecture | ANN Acc. (%) | SNN Acc. (%) | $T_w$=16 SNN Acc. (%) | $T_w$=16 $\Delta$ Acc. (%) | $T_w$=32 SNN Acc. (%) | $T_w$=32 $\Delta$ Acc. (%) |
|---|---|---|---|---|---|---|---|---|---|
| CIFAR-10 | PTL [57] | IF | VGG-11 | 90.59 | - | 91.24 | 0.65 | - | - |
| | TET [13] | LIF | VGG-11* | - | 92.98 ($T_w$=6) | - | - | - | - |
| | Offline (Online) LTL | LIF | VGG-11 | 92.89 | - | 93.20 (92.11) | 0.31 (-0.78) | 93.29 (92.75) | 0.40 (-0.14) |
| | Offline (Online) LTL | IF | VGG-11 | 92.89 | - | 93.08 (92.03) | 0.19 (-0.86) | 93.25 (92.31) | 0.36 (-0.58) |
| | SPIKE-NORM [49] | IF | VGG-16 | 91.70 | 91.55 ($T_w$=2500) | - | - | - | - |
| | Hybrid Train [46] | IF | VGG-16 | 92.81 | 91.13 ($T_w$=100) | - | - | - | - |
| | Calibration [34] | IF | VGG-16 | 95.72 | - | - | - | 93.71 | -2.01 |
| | RMP [22] | IF | VGG-16 | 93.63 | 93.63 ($T_w$=2048) | - | - | - | - |
| | TSC [21] | IF | VGG-16 | 93.63 | 93.63 ($T_w$=2048) | - | - | - | - |
| | Opt. [12] | IF | VGG-16 | 92.34 | - | 92.29 | -0.05 | 92.29 | -0.05 |
| | Opt. mem. [4] | IF | VGG-16 | 94.57 | - | 93.38 | -1.19 | 94.20 | -0.37 |
| | QCFS [5] | IF | VGG-16 | 95.52 | - | 95.40 | -0.12 | 95.54 | 0.02 |
| | Diet-SNN [45] | LIF | VGG-16 | - | 92.70 ($T_w$=5) | - | - | - | - |
| | TET [13] | LIF | VGG-16* | - | 93.49 ($T_w$=6) | - | - | - | - |
| | Offline (Online) LTL | LIF | VGG-16 | 94.05 | - | 93.23 (92.85) | -0.82 (-1.20) | 93.92 (93.52) | -0.13 (-0.53) |
| | Offline (Online) LTL | IF | VGG-16 | 94.05 | - | 93.04 (92.22) | -1.01 (-1.83) | 93.72 (93.17) | -0.33 (-0.88) |
| | SPIKE-NORM [49] | IF | ResNet-20 | 89.10 | 87.46 ($T_w$=25000) | - | - | - | - |
| | Hybrid Train [46] | IF | ResNet-20 | 93.15 | 92.22 ($T_w$=250) | - | - | - | - |
| | Calibration [34] | IF | ResNet-20 | 95.46 | - | - | - | 94.78 | -0.68 |
| | RMP [22] | IF | ResNet-20 | 91.47 | 91.36 ($T_w$=2048) | - | - | - | - |
| | TSC [21] | IF | ResNet-20 | 91.47 | 91.42 ($T_w$=2048) | - | - | - | - |
| | Opt. [12] | IF | ResNet-20 | 93.61 | - | 92.41 | -1.20 | 93.30 | -0.31 |
| | Opt. mem. [4] | IF | ResNet-20 | 92.74 | - | 87.22 | -5.52 | 91.88 | -0.86 |
| | QCFS [5] | IF | ResNet-20 | 91.77 | - | 91.62 | -0.15 | 92.24 | 0.47 |
| | Diet-SNN [45] | LIF | ResNet-20 | - | 92.54 ($T_w$=10) | - | - | - | - |
| | Offline (Online) LTL | LIF | ResNet-20 | 95.36 | - | 94.76 (93.15) | -0.60 (-2.21) | 95.25 (94.95) | -0.11 (-0.41) |
| | Offline (Online) LTL | IF | ResNet-20 | 95.36 | - | 94.82 (91.33) | -0.54 (-4.03) | 95.28 (93.84) | -0.08 (-1.52) |
| CIFAR-100 | Hybrid Train [46] | IF | VGG-11 | 71.21 | 67.87 ($T_w$=125) | - | - | - | - |
| | Offline (Online) LTL | LIF | VGG-11 | 71.59 | - | 72.63 (70.80) | 1.04 (-0.79) | 73.08 (72.45) | 1.49 (0.86) |
| | Offline (Online) LTL | IF | VGG-11 | 71.59 | - | 72.74 (70.81) | 1.15 (-0.78) | 72.71 (72.29) | 1.12 (0.70) |
| | SPIKE-NORM [49] | IF | VGG-16 | 71.22 | 70.77 ($T_w$=2500) | - | - | - | - |
| | Calibration [34] | IF | VGG-16 | 77.89 | - | - | - | 73.55 | -4.34 |
| | RMP [22] | IF | VGG-16 | 71.22 | 70.93 ($T_w$=2048) | - | - | - | - |
| | TSC [21] | IF | VGG-16 | 71.22 | 70.97 ($T_w$=2048) | - | - | - | - |
| | Opt. [12] | IF | VGG-16 | 70.49 | - | 65.94 | -4.55 | 69.80 | -0.69 |
| | Opt. mem. [4] | IF | VGG-16 | 76.31 | - | 70.72 | -5.59 | 74.82 | -1.49 |
| | QCFS [5] | IF | VGG-16 | 76.28 | - | 76.24 | -0.04 | 77.01 | 0.73 |
| | Diet-SNN [45] | LIF | VGG-16 | - | 69.67 ($T_w$=5) | - | - | - | - |
| | TET [13] | LIF | VGG-16* | - | 72.45 ($T_w$=6) | - | - | - | - |
| | Offline (Online) LTL | LIF | VGG-16 | 74.42 | - | 74.19 (71.09) | -0.23 (-3.33) | 74.92 (72.97) | 0.50 (-1.45) |
| | Offline (Online) LTL | IF | VGG-16 | 74.42 | - | 73.67 (69.55) | -0.75 (-4.87) | 74.56 (71.34) | 0.14 (-3.08) |
| | SPIKE-NORM [49] | IF | ResNet-20 | 69.72 | 64.09 ($T_w$=2500) | - | - | - | - |
| | Calibration [34] | IF | ResNet-20 | 77.16 | - | - | - | 76.32 | -0.84 |
| | RMP [22] | IF | ResNet-20 | 68.72 | 67.82 ($T_w$=2048) | - | - | - | - |
| | TSC [21] | IF | ResNet-20 | 68.72 | 68.18 ($T_w$=2048) | - | - | - | - |
| | Opt. [12] | IF | ResNet-20 | 69.80 | - | 63.73 | -6.07 | 68.40 | -1.40 |
| | Opt. mem. [4] | IF | ResNet-20 | 70.43 | - | 52.34 | -18.09 | 67.18 | -3.25 |
| | QCFS [5] | IF | ResNet-20 | 69.94 | - | 67.33 | -2.61 | 69.82 | -0.12 |
| | Diet-SNN [45] | LIF | ResNet-20 | – | 64.07 ($T_w$=5) | - | - | - | - |
| | Offline (Online) LTL | LIF | ResNet-20 | 76.36 | - | 75.62 (73.22) | -0.74 (-3.14) | 76.12 (75.02) | -0.24 (-1.34) |
| | Offline (Online) LTL | IF | ResNet-20 | 76.36 | - | 75.22 (71.16) | -1.14 (-5.20) | 76.08 (73.93) | -0.28 (-2.43) |
| Tiny-ImageNet | DCT [17] | IF | VGG-13 | 56.90 | 52.43 ($T_w$=125) | - | - | - | - |
| | Offline (Online) LTL | LIF | VGG-13 | 56.16 | - | 55.37 (54.82) | -0.79 (-1.34) | 55.85 (55.91) | -0.31 (-0.25) |
| | Offline (Online) LTL | IF | VGG-13 | 56.16 | - | 55.27 (54.28) | -0.89 (-1.88) | 55.85 (55.61) | -0.31 (-0.55) |
| | Hybrid [31] | IF | VGG-16 | 56.56 | 51.92 ($T_w$=150) | - | - | - | - |
| | Calibration [34] | IF | VGG-16* | 60.95 | - | 44.39 | -16.56 | 53.96 | -6.99 |
| | QCFS [5] | IF | VGG-16* | 57.82 | - | 44.53 | -13.29 | 53.54 | -4.28 |
| | Offline (Online) LTL | LIF | VGG-16 | 57.39 | - | 56.85 (56.87) | -0.54 (-0.52) | 57.73 (57.45) | 0.34 (0.06) |
| | Offline (Online) LTL | IF | VGG-16 | 57.39 | - | 56.59 (56.39) | -0.80 (-1.00) | 57.49 (57.00) | 0.10 (-0.39) |
| | Calibration [34] | IF | ResNet-20* | 58.29 | - | 44.89 | -13.40 | 52.79 | -5.5 |
| | QCFS [5] | IF | ResNet-20* | 55.81 | - | 52.32 | -3.49 | 55.06 | -0.75 |
| | Offline (Online) LTL | LIF | ResNet-20 | 56.68 | - | 56.24 (52.52) | -0.44 (-4.16) | 57.42 (55.32) | 0.74 (-1.36) |
| | Offline (Online) LTL | IF | ResNet-20 | 56.68 | - | 56.28 (51.92) | -0.40 (-4.76) | 56.48 (53.92) | -0.20 (-2.76) |

\* Our reproduced results using publicly available codes.

maintain a high accuracy even at $T_w = 4$. It suggests the surrogate gradient approach adopted in SNN training can outperform the naive straight-through estimator [25] in addressing the discontinuity in the activation function. Moreover, the offline LTL rule performs better than the STBP rule, which can be explained by the fact that the gradient approximation errors of the surrogate gradient function will not compound across layers as in the STBP training [57]. Similar results have also been observed in the VGG-16 architecture (See Supplementary Materials Section E).

To shed light on the energy efficiency of LTL trained SNN models, we follow the common practice by computing the total synaptic operations (SynOps) ratio of SNN to ANN under different $T_w$ [56]. In general, the total SynOps (i.e., FLOPs) required by an ANN is a constant number that depends on the specific network architecture in use, while the total computations typically increase linearly with $T_w$ for rate-based SNNs. As shown in Figure 2(b), to achieve comparable accuracy to the ANN counterpart, our SNN only requires $0.43\times$ total SynOps with $T_w = 8$. It is worth noting that the multiply-accumulate (MAC) operation required by ANNs is $5\times$ more expensive than the accumulate

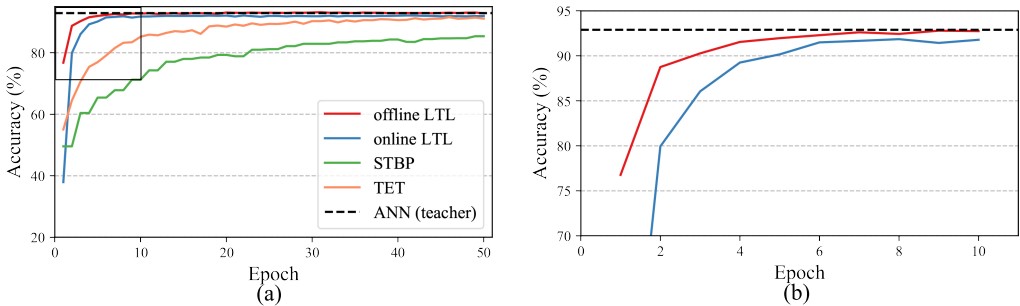

Figure 3: Comparison of the learning curves of offline LTL, online LTL, STBP, and TET learning rules. The experiments are performed on the CIFAR-10 dataset with the VGG-11 architecture.

(AC) operations used in SNNs under 45 nm CMOS technology [23]. As such, our SNNs can yield an order of magnitude energy saving over the mainstream ANNs. Notably, this value can be further boosted when deploying our SNNs onto the emerging mixed-signal NC chips. Moreover, when comparing with QCFS [5] which is a recently introduced ANN-to-SNN conversion method, the proposed LTL rule requires fewer SynOps probably due to a higher percentage of neurons remaining silent in our network. This can be explained by the round-down operation that has been taken when calculating the target spike count in our method.

### 3.3 Rapid Learning Convergence

By decoupling the learning of each layer through local objectives, the proposed LTL rule cleverly circumvents the spatial credit assignment and reduces the credit assignment to only the temporal domain. Moreover, the gradient approximation errors of the surrogate gradient learning method are alleviated as the effective gradient transportation distance is reduced. Taking benefits from these, the network convergence speed can be improved significantly compared to those trained with the STBP and TET [13] rule. As the learning curves are shown in Figure 3, both the offline and online LTL rules can converge rapidly within five epochs on the CIFAR-10 dataset. In contrast, the accuracy is still increasing at Epoch 50 for both the STBP and TET learning rules. Note that we follow the same settings for all these experiments, except for the learning objectives. Similar results have also been observed in the VGG-16 architecture (see Supplementary Materials Section F).

### 3.4 Low Memory and Time Complexity

In this section, we study the memory and time complexity of the proposed LTL rule and compare it against other popular learning rules in Table 2. The STBP [58] rule requires storing the intermediate state of each neuron for all time steps and layers so as to be used during backpropagation, resulting in $\mathcal{O}(MT_wL)$ memory complexity, where $M$ is the number of neurons in a layer, $T_w$ is the time window size, and $L$ is the total number of layers. The offline LTL rule has the same memory complexity as the STBP rule if all layers are trained in parallel. However, as will be discussed

Table 2: Comparison of the memory and time complexity of different learning rules. $N$: number of input neurons to a layer, $M$: number of neurons in a layer, $N_r$: number of readout neurons in a layer, $L$: number of layers, $T_w$: time window size

| Method | Memory | Time |
|---|---|---|
| STBP [58] | $\mathcal{O}(MT_wL)$ | $\mathcal{O}(NMT_wL)$ |
| DECOLLE [26] | $\mathcal{O}(1)$ | $\mathcal{O}(NM + MN_r)$ |
| Offline LTL | $\mathcal{O}(MT_wL)$ | $\mathcal{O}(NMT_w)$ |
| Online LTL | $\mathcal{O}(1)$ | $\mathcal{O}(NM)$ |

soon, we can reduce the memory complexity by training all layers in sequence. Notably, by updating the network parameters using only local information, the online LTL and DECOLLE [26] learning rules do not require storing any intermediate state, resulting in $\mathcal{O}(1)$ memory complexity.

In terms of time complexity, each parameter update of the STBP rule requires $NMT_w$ multiplications, where $N$ is the number of input neurons to a layer. Due to layer-wise interlocking, all layers need to be updated in sequence from top to bottom, requiring $\mathcal{O}(NMT_wL)$ time complexity to update

all network parameters once. For offline LTL, by decoupling the dependency between layers and training all layers in parallel, the time complexity can be reduced to $\mathcal{O}(NMT_w)$. As mentioned earlier, we can trade off the memory complexity with the time complexity by training the network layers progressively one after the other. In this way, the memory usage of the offline LTL rule will be reduced to $\mathcal{O}(MT_w)$, while the time consumption will be increased to $\mathcal{O}(NMT_wL)$. To validate the effectiveness of this training strategy, we perform an experiment on the CIFAR-10 dataset using VGG-11 architecture and we obtain a comparable result (92.98%) to that obtained when all layers are trained concurrently (93.20%). In this work, to achieve the shortest training time, we use the parallel training strategy as the default for all the other reported experiments. Other users can flexibly select the training strategy according to their available computing resources.

For online LTL, by leveraging the temporal local errors, the time complexity can be further reduced to $\mathcal{O}(NM)$. Note that the DECOLLE requires an additional readout layer to calculate local errors, thereby requiring more time than online LTL. It is also worth mentioning that by gating the parameter updates with the proposed efficient on-chip implementation, we can further reduce the time complexity for online LTL. To provide a more concrete analysis of the memory and time complexity, we have measured the actual GPU memory usage and the training time on the CIFAR-10 dataset, which aligns well with the earlier theoretical analysis (see Supplementary Materials Section G). In sum, by coupling the low computational complexity with the rapid learning convergence discussed in Section 3.3, the proposed LTL rule can dramatically improve the training efficiency of SNNs.

### 3.5 Robust to Hardware-related Noises

Device non-ideality issues of analog computing substrates remain a major hurdle for deploying SNNs on ultra-low-power mixed-signal NC chips. The proposed LTL rule can, however, address this challenge by enabling on-chip training in a noise-aware manner. Here, we examine the effectiveness of the on-chip LTL rule in addressing hardware-related noises and we focus on four typical noise sources: device mismatch, quantization noise, thermal noise, and neuron silencing [6]. In particular, we use the noise models introduced by Julian et al. [6], which were measured on DYNAP-SE [36] mixed-signal neuromorphic processor. Therefore, it can faithfully reflect the actual hardware noises that one can expect in the real scenario. To understand the effect of these noises, we first take an LTL pre-trained SNN and apply noise at different levels to it. Then, we use the online LTL rule to calibrate the pre-trained SNN model under the given noise.

**Device Mismatch** Device mismatch causes neuronal and synaptic parameters to vary from the desired ones. To simulate the parameter mismatch, we add Gaussian noise to both the initial network parameters and their parameter updates following $\theta' \sim \mathcal{N}(\theta, \sigma\theta)$. In this model, $\sigma$ controls the level of mismatch, and we vary it from 0.05 to 0.4 to simulate different levels of noise. As shown in Figure 4(a), the accuracy of the pre-trained SNN degrades by more than 20% under the mismatch level $\sigma = 0.4$. Notably, the proposed LTL rule can perform fast calibration and restore the original accuracy with only one training epoch. This will be advantageous for those devices that have poor endurance.

**Quantization Noise** The non-volatile memory devices typically have a limited number of analog states, thereby limiting the usable number of bits per network parameter. To simulate such a quantization noise, we perform post-training-quantization to the bit-width of $7, 6, 5, 4, 3$. As shown in Figure 4(b), the SNN performs robustly to the quantization noise for bit-precision above 3. After which, the accuracy drops sharply by more than 30%. Following the quantization aware training [25], the proposed LTL rule can alleviate the effect of this type of noise and successfully recover the accuracy by around 17%. Nevertheless, due to the reduced representation power, the resulting 3-bit precision model is still lagging behind the full precision one by about 15%.

**Thermal Noise** Thermal noise is intrinsic to the neuromorphic devices, which can be modeled as Gaussian noise on the input currents $I' \sim \mathcal{N}(I, \sigma I)$. Similar to the device mismatch noise, we simulate different levels of thermal noise by varying the $\sigma$ from 0.01 to 0.2. As illustrated in Figure 4(c), the model is highly sensitive to thermal noise and the accuracy drops steadily with a growing level of noise. We notice that the LTL rule can address such noise and quickly recover the accuracy in large part with only one training epoch.

**Neuron Silencing** Neuron silencing noise corresponds to the situation when a portion of spiking neurons fails to respond, and hence disturbs the network dynamics. We simulate neuron silencing

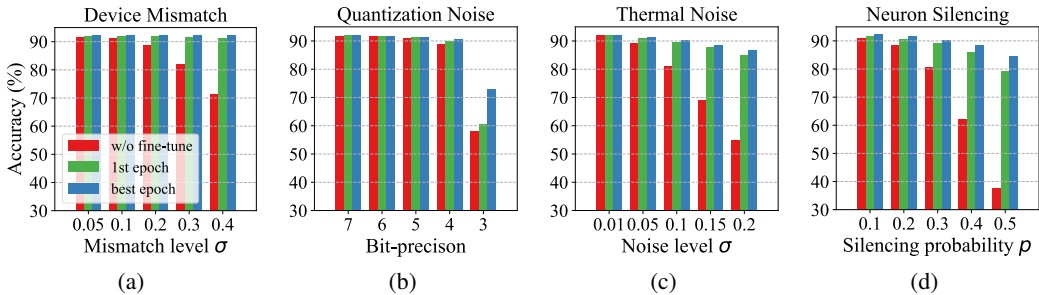

Figure 4: The classification accuracy is a function of the level of device noises for (a) device mismatch, (b) quantization noise, (c) thermal noise, and (d) neuron silencing. w/o fine-tune: without any noise-aware fine-tuning, 1st epoch: the result after training with the online LTL rule for 1 epoch, best epoch: the best result obtained when training with the online LTL rule for 50 epochs. The experiments are performed on the CIFAR-10 dataset with the VGG-11 architecture.

noise by randomly masking the neuron outputs with a failure rate varying from 10% to 50%. As shown in Figure 4(d), the accuracy drops steadily with a growing amount of failed neurons. Similar to other noise types, the accuracies are quickly restored with the on-chip LTL training and maintained above 80% even under the challenging case when 50% of the neurons failed.

**Comparison with the HIL Training Approach** The proposed LTL rule exhibits better noise robustness and scalability than the HIL training approaches. To demonstrate this, we compare the online LTL rule with the HIL training approach [10] in addressing the aforementioned device-related noises (see Supplementary Material Section H for the details of experimental set-up and results). The LTL rule achieves a comparable accuracy with the HIL approach for small and moderate levels of noise (see Table 7), but it appears to be more robust when facing stronger noise levels (see Table 8). Additionally, the LTL rule is more robust to the choice of the learning rate (see Table 9). In summary, the proposed LTL learning rule is both effective and efficient for addressing the device non-idealities of mixed-signal NC chips.

## 4    Conclusion

In this paper, we propose the Local Tandem Learning (LTL) rule for SNNs. Taking inspiration from the teacher-student learning approach, we leverage the highly effective feature representations of easily trainable ANN to guide the training of SNNs. This simplifies the credit assignment problem and leads to rapid network convergence with significantly reduced computational complexity. We also demonstrate the required temporal credit assignment can be well approximated with locally available information, and we provide an efficient on-chip implementation of the proposed LTL rule. Under this on-chip setting, we demonstrate that the LTL rule can provide robustness against various device non-ideality issues. It, therefore, opens up myriad opportunities for rapid and efficient training and deployment of SNNs on ultra-low-power mix-signal NC platforms. For future work, we will explore the proposed teacher-student learning approach on the recurrent neural networks and transformer architectures so as to handle those temporally rich signals, such as speech, video, and text.

## Acknowledgments and Disclosure of Funding

This research work is supported by IAF, A*STAR, SOITEC, NXP, National University of Singapore under FD-fAbrICS: Joint Lab for FD-SOI Always-on Intelligent & Connected Systems (Award I2001E0053), and National Research Foundation, Singapore under its Medium Sized Centre for Advanced Robotics Technology Innovation (WBS: A-0009428-09-00). This research is also supported by National Science Foundation of China (Grant Number: 62106038), Guangdong Provincial Key Laboratory of Big Data Computing, The Chinese University of Hong Kong, Shenzhen, China (Grant No. B10120210117-KP02), Shenzhen Research Institute of Big Data, the National Key Research and Development Program of China (Grant No. 2021ZD0200300), CCF-Hikvision Open Fund, and The Hong Kong Polytechnic University under Grant No. P0043563.

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
