# Supplementary Materials

## A    Derivation for gradients

### A.1    Offline Learning

In the following, we will derive the gradients for $t = T_w$ and $t < T_w$ separately. To derive the gradients to the weight and bias terms, we first define

$$\delta_i^l[t] = \frac{\partial \mathcal{L}^l}{\partial S_i^l[t]}. \tag{12}$$

1. For $t = T_w$:

The $\delta_i^l[T_w]$ can be directly calculated from the layer-wise local loss function, i.e., Eq. (4):

$$\delta_i^l[T_w] = \frac{\partial \mathcal{L}^l}{\partial S_i^l[T_w]} = -\frac{2}{T_w} \left( \frac{\hat{y}_i^l}{y_{norm}^l} - \frac{1}{T_w} \Sigma_{k=1}^{T_w} S_i^l[k] \right) \tag{13}$$

With this, we can obtain

$$\frac{\partial \mathcal{L}^l}{\partial U_i^l[T_w]} = \frac{\partial \mathcal{L}^l}{\partial S_i^l[T_w]} \frac{\partial S_i^l[T_w]}{\partial U_i^l[T_w]} = \delta_i^l[T_w] \frac{\partial S_i^l[T_w]}{\partial U_i^l[T_w]} \tag{14}$$

2. For $t < T_w$:

The $\delta_i^l[t]$ is obtained following the error backpropagation through time algorithm. By unrolling the neuronal states along the temporal domain and applying the chain rule, we have

$$\delta_i^l[t] = \frac{\partial \mathcal{L}^l}{\partial S_i^l[t]} = \frac{\partial \mathcal{L}^l}{\partial S_i^l[t+1]} \frac{\partial S_i^l[t+1]}{\partial U_i^l[t+1]} \frac{\partial U_i^l[t+1]}{\partial S_i^l[t]} + \frac{\partial \mathcal{L}^l}{\partial S_i^l[T_w]} = \delta_i^l[t+1] \frac{\partial S_i^l[t+1]}{\partial U_i^l[t+1]}(-\vartheta) + \delta_i^l[T_w] \tag{15}$$

We can further obtain $\frac{\partial \mathcal{L}^l}{\partial U_i^l[t]}$ as

$$\frac{\partial \mathcal{L}^l}{\partial U_i^l[t]} = \frac{\partial \mathcal{L}^l}{\partial S_i^l[t+1]} \frac{\partial S_i^l[t+1]}{\partial U_i^l[t+1]} \frac{\partial U_i^l[t+1]}{\partial U_i^l[t]} + \frac{\partial \mathcal{L}^l}{\partial S_i^l[t]} \frac{\partial S_i^l[t]}{\partial U_i^l[t]} = \delta_i^l[t+1] \frac{\partial S_i^l[t+1]}{\partial U_i^l[t+1]}\alpha + \delta_i^l[t] \frac{\partial S_i^l[t]}{\partial U_i^l[t]} \tag{16}$$

Finally, the gradients for network weight and bias terms can be computed by substituting Eqs. (14) and (16) into the following equations:

$$\frac{\partial \mathcal{L}^l}{\partial w_{ij}^l} = \sum_t^{T_w} \frac{\partial \mathcal{L}^l}{\partial U_i^l[t]} \frac{\partial U_i^l[t]}{\partial w_{ij}^l} = \sum_t^{T_w} \frac{\partial \mathcal{L}^l}{\partial U_i^l[t]} S_j^{l-1}[t-1] \tag{17}$$

$$\frac{\partial \mathcal{L}^l}{\partial b_i^l} = \sum_t^{T_w} \frac{\partial \mathcal{L}^l}{\partial U_i^l[t]} \frac{\partial U_i^l[t]}{\partial b_i^l} = \sum_t^{T_w} \frac{\partial \mathcal{L}^l}{\partial U_i^l[t]} \tag{18}$$

## B    Experimental details

### B.1    Datasets

**CIFAR-10** [29] This dataset contains 60,000 colored images from 10 classes. Each of the images with the size of $32 \times 32 \times 3$. All the images are split into 50,000 and 10,000 for training and testing, respectively.

**CIFAR-100** [29] This dataset contains 60,000 colored images from 100 classes. Each of the images with the size of $32 \times 32 \times 3$. All the images are split into 50,000 and 10,000 for training and testing, respectively.

**Tiny ImageNet** [55] This dataset contains 110,000 colored images from 200 classes. Each of the images with the size of $64 \times 64 \times 3$. All the images are split into 100,000 and 10,000 for training and testing, respectively.

For all the datasets, we follow the similar data pre-processing techniques used in [17], including resize and random crop, random horizontal flip, and data normalization. More details can be found in our released code.

### B.2 Hyper-parameters for SNN

We fine-tuned the SNN hyper-parameters for different datasets as presented in Table 3.

Table 3: Hyper-parameters setting. $\tau_m$: membrane time constant of LIF neuron ($\tau_m = \infty$ for IF neuron), $\vartheta$: neuronal firing threshold, and $p$: permission range of membrane potential that allows gradients to pass through.

| Dataset | $\tau_m$ | $\vartheta$ | $p$ |
|---------|-----|-----|-----|
| CIFAR-10 | 10 | 0.6 | 0.4 |
| CIFAR-100 | 10 | 0.5 | 0.4 |
| Tiny-ImageNet | 10 | 0.5 | 0.4 |

### B.3 Network architecture and training configuration

To facilitate comparison with other work, we perform experiments with VGG-11, 13, and 16, whose network architectures are summarized as the following.

Table 4: Summary of network architectures. $n$C3: convolutional layer with $n$ output channels, $3 \times 3$ kernel size, and stride of 2. $n$FC: linear layer with $n$ output features.

| Network | Architecture |
|---------|-------------|
| VGG-11 | Input-64C3-128C3S2-256C3-256C3S2-512C3-512C3S2-512C3-512C3C2-512FC-512FC-Classes |
| VGG-13 | Input-64C3-64C3-128C3-128C3S2-256C3-256C3S2-512C3-512C3S2-512C3-512C3C2-4096FC-4096FC-Classes |
| VGG-16 | Input-64C3-64C3-128C3-128C3S2-256C3-256C3-256C3S2-512C3-512C3-512C3S2-512C3-512C3-512C3C2-4096FC-4096FC-Classes |

For VGG, the pooling layers are replaced with convolutional layers that have a stride of 2, and the dropout is applied after fully connected (FC) layers. We use the Pytorch library to accelerate training with multi-GPU machines. We train all teacher ANNs for 200 epochs using an SGD optimizer with a momentum of 0.9 and weight decay of $5e^{-4}$. The initial learning rates are set to 0.01, 0.01, and 0.1 for CIFAR-10, Tiny-ImageNet, and CIFAR-100 datasets, respectively; the learning rates decay by 10 at 60, 120, and 160 epochs. For the LTL training stage, we train the student SNNs with the Adam optimizer for 100, 50, and 50 epochs for CIFAR-10, CIFAR-100, Tiny-ImageNet, respectively. The initial learning rate is set to $1e^{-4}$ for all SNNs and decays its value by 5 every 10 epochs for CIFAR-10 and 5 epochs for CIFAR-100 and Tiny-ImageNet, respectively. We train all the models on Nvidia Geforce GTX 1080Ti GPUs with 12 GB memory for $T_w = 16$ and below, and we use the GPU cluster that has GPUs with 40 GB memory for $T_w = 32$.

## C Study of the warm-up period on the online learning performance

To study the effect of the warm-up period $T_{warm}$ on the online learning performance, we perform a study by progressively increasing $T_{warm}$, during which the parameter updates are not allowed. The experiments are conducted on CIFAR-10 dataset with VGG-16 and $T_w = 16$. As the results reported in Table 5, it is clear that increasing $T_{warm}$ will lead to better gradient approximation as evidenced by the improved test accuracy.

Table 5: Ablation study on the warm-up time steps $T_{warm}$.

| $T_{warm}$(online) | SNN(LIF) | SNN(IF) |
|---|---|---|
| 2 | 91.07 | 90.38 |
| 4 | 92.27 | 92.20 |
| 6 | 92.54 | 92.37 |
| 8 | 92.71 | 92.26 |
| 10 | 92.73 | 92.45 |
| 12 | 92.74 | 92.15 |
| 14 | 93.00 | 92.60 |

## D Online LTL rule perform as effectively as the offline version

To further shed light on why the online LTL rule can perform as effectively as the offline version, we conduct an experiment to analyze the degree of mismatch between their calculated gradients. In this experiment, we first pre-train a teacher ANN model on the MNIST dataset [33] with a 4-layer MLP architecture (i.e., 784-800-800-800-10). Then, we randomly initialize a student SNN model and draw 50 random batches of 128 samples to calculate the gradients at each layer. As shown in Figure 5, the cosine similarities between offline and online calculated gradients remain higher than 0.86 for all the hidden layers. According to the hyperdimensional computing theory [27], any two high dimensional random vectors are approximately orthogonal. It suggests the online estimated gradients are very close to the offline calculated ground truth values and guarantees that the desired learning dynamics can be well approximated.

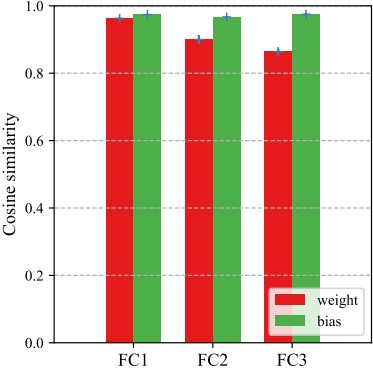

Figure 5: Cosine similarities between the online and offline calculated gradients on a MLP architecture. The error bar indicates the standard deviation across 50 random data batches.

## E Rapid and efficient pattern recognition on VGG-16

We show here that similar results as described in Section 3.2 can be also obtained on VGG-16. As shown in Figure 6(a), both LTL and STBP trained SNNs can maintain a high accuracy even with an extremely short time window (i.e., $T_w = 4$), while the quantized ANN degrades significantly when the quantization level is below 16. In addition, our VGG-16 can achieve competitive accuracy with only $0.57\times$ total SynOps as compared to its analog counterpart as shown in Figure 6(b).

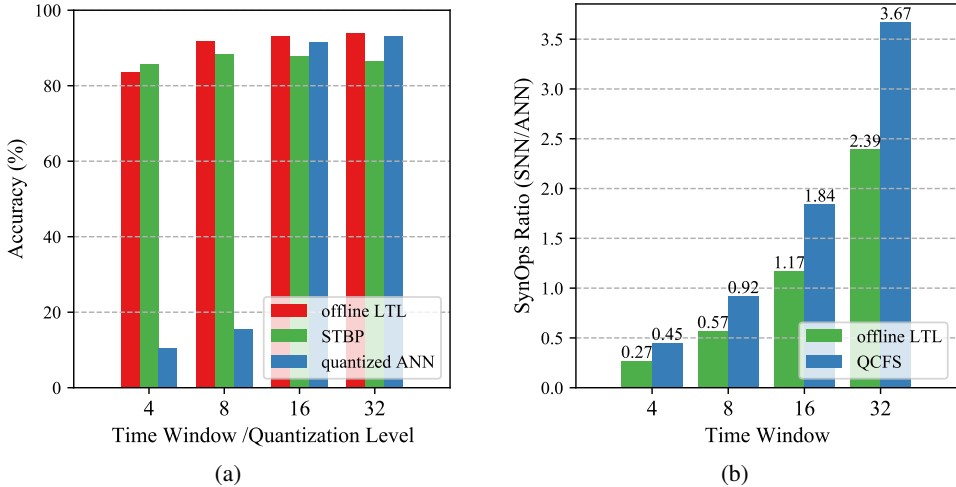

(a)                                                                                    (b)

Figure 6: (a) Comparison of SNN and ANN accuracies under different time windows (SNN) and quantization levels (ANN). The results are obtained on the CIFAR-10 dataset using the VGG-16 architecture. (b) The ratio of total synaptic operations between SNN and ANN as a function of the time window.

# F   Rapid network convergence for VGG-16 architectures

We show here that a similar result of the experiment described in Section 3.3 can be obtained on the VGG-16 architecture. Both the offline and online LTL rules converge rapidly within 5 epochs on the CIFAR-10 dataset, which is much faster than the baseline STBP and TET rules.

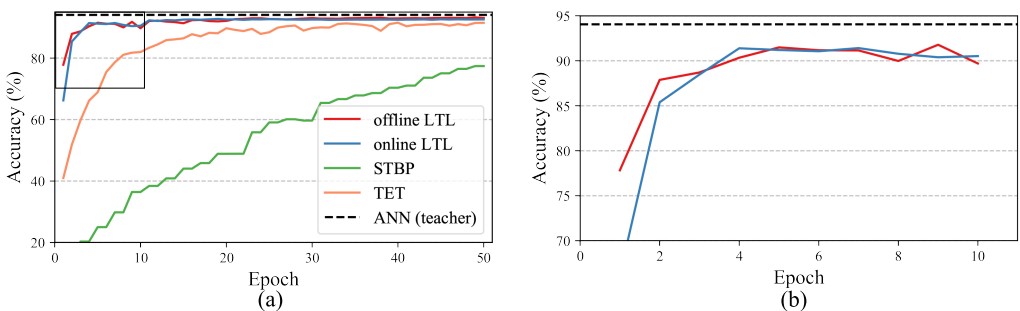

(a)                                                                                    (b)

Figure 7: Comparison of the learning curves of offline LTL, online LTL, STBP and TET learning rules. The experiments are performed on the CIFAR-10 dataset with the VGG-16 architecture.

# G   Empirical analysis on memory and time complexity

We measure the actual memory consumption and training time using the VGG11 architecture and CIFAR-10 dataset. We present the average results recorded over 10 training epochs in the Table below. In general, the GPU memory usage scales up almost linearly with the time window size (i.e., T=16) for the STBP and Offline LTL methods, which follows our theoretical analysis. Compared to the STBP rule, the offline LTL rule requires slightly more memory space for the calculation of the layer-wise loss functions. As for per epoch training time, although both offline and online LTL rules are significantly faster than STBP rule, the ratios of speed up scale poorer than the theoretical ones. We would like to acknowledge that we adopt the primitive and unoptimized Pytorch GPU kernels in our implementation. To achieve the desired theoretical speed up, it requires customized GPU kernels that can perform the training both in parallel across time and layers and we leave this as future work.

Table 6: Comparison of the empirical memory and time complexity of different learning rules.

| Method | GPU Memory Usage (MB) | Training Time per Epoch (s) |
|---|---|---|
| STBP | 7072 | 489.80 |
| Offline LTL | 7472 | 389.32 |
| Online LTL | 718 | 103.30 |

# H Comparison with hardware-in-the-loop training approach

We compare the online LTL rule with the hardware-in-the-loop (HIL) training approach in addressing device-related noises introduced in Section 3.5. Table 7 reports the fine-tuned accuracies under different noise levels with a pre-trained SNN of $92.11\%$ test accuracy. These results suggest both approaches can achieve comparable performance in addressing the low and moderate levels of device noises.

Table 7: Comparison of the test accuracies of online LTL and HIL training rules on CIFAR-10 dataset.

| Noise Type | Noise Level | Before Fine-tuning | LTL Fine-tuned | HIL Fine-tuned |
|---|---|---|---|---|
| Device mismatch | $\sigma = 0.05$ | 91.67 | 92.22 | 92.66 |
| | $\sigma = 0.10$ | 91.26 | 92.23 | 92.23 |
| | $\sigma = 0.20$ | 88.63 | 92.21 | 90.71 |
| | $\sigma = 0.30$ | 81.96 | 92.29 | 87.55 |
| | $\sigma = 0.40$ | 71.20 | 92.26 | 82.11 |
| Quantization noise | 7 bits | 91.83 | 92.08 | 91.58 |
| | 6 bits | 91.71 | 91.82 | 91.43 |
| | 5 bits | 91.07 | 91.51 | 90.48 |
| | 4 bits | 88.92 | 90.64 | 89.37 |
| | 3 bits | 58.22 | 73.07 | 55.50 |
| Thermal noise | $\sigma = 0.01$ | 91.94 | 92.20 | 92.42 |
| | $\sigma = 0.05$ | 89.25 | 91.36 | 91.98 |
| | $\sigma = 0.10$ | 81.20 | 90.23 | 90.40 |
| | $\sigma = 0.15$ | 69.18 | 88.51 | 89.09 |
| | $\sigma = 0.20$ | 55.03 | 86.77 | 86.92 |
| Neuron silence | $p = 0.1$ | 90.96 | 92.14 | 92.27 |
| | $p = 0.2$ | 88.32 | 91.45 | 92.37 |
| | $p = 0.3$ | 80.58 | 90.28 | 92.03 |
| | $p = 0.4$ | 62.13 | 88.42 | 91.46 |
| | $p = 0.5$ | 37.38 | 84.45 | 90.54 |

To demonstrate the proposed layer-wise training approach can achieve better noise robustness and scalability than the HIL training approach, we further increase the level of device noise and tested the pre-trained SNNs on the MNIST dataset. As the results summarised in Table 8, the LTL rule is highly robust to different levels of noise and can also scale up freely to deeper VGG-16 architecture. In contrast, the performance degrades significantly for the HIL learning method with increasing levels of noise and network depth. This can be explained by the fact that the gradients estimated at each layer tend to be noisy and the errors accumulated across layers during training. Whereas our layer-wise training approach can effectively overcome this problem.

Table 8: Comparison of the test accuracies of online LTL and HIL training rules on CIFAR-10 dataset with VGG-9 and VGG-11 architectures.

| Architecture | Pre-trained Acc. | Noise Level | Before Fine-tuning | LTL Fine-tuned | HIL fine-tuned |
|---|---|---|---|---|---|
| VGG-9 | 99.45 | $\sigma = 0.5$ | 99.00 | 99.42 | 99.02 |
| | | $\sigma = 1.0$ | 95.99 | 99.36 | 97.31 |
| | | $\sigma = 1.5$ | 70.51 | 99.34 | 78.40 |
| VGG-11 | 99.55 | $\sigma = 0.5$ | 98.29 | 99.69 | 99.40 |
| | | $\sigma = 1.0$ | 87.63 | 99.60 | 86.52 |
| | | $\sigma = 1.5$ | 39.21 | 99.52 | 10.42 |

It is beneficial if the learning rules are robust to the choice of hyper-parameters. To further investigate on this perspective, we compare the performance of the online LTL and HIL rules at different learning rates. As shown in Table 9, the LTL rule can tolerate a larger learning rate than the HIL rule.

Table 9: Comparison of the test accuracies of LTL (HIL) rule at different learning rates. The experiments are conducted on CIFAR-10 dataset with VGG-11. **lr:** learning rate.

| Noise Type | Noise Level | $lr = 0.0001$ | $lr = 0.001$ | $lr = 0.01$ |
|---|---|---|---|---|
| Device mismatch | $\sigma = 0.05$ | 92.05 (90.87) | 91.29 (10.00) | 74.09 (10.00) |
| | $\sigma = 0.10$ | 92.06 (92.59) | 91.34 (10.00) | 72.96 (10.00) |
| | $\sigma = 0.20$ | 92.09 (91.89) | 91.69 (10.00) | 10.00 (10.00) |
| | $\sigma = 0.30$ | 92.22 (89.61) | 91.85 (10.00) | 77.83 (10.00) |
| | $\sigma = 0.40$ | 92.43 (86.02) | 92.09 (10.00) | 84.75 (10.00) |
| Quantization noise | 7 bits | 92.35 (91.25) | 92.33 (89.87) | 92.38 (35.66) |
| | 6 bits | 91.84 (91.24) | 91.84 (90.22) | 91.98 (75.04) |
| | 5 bits | 91.81 (90.40) | 91.96 (87.91) | 91.84 (70.99) |
| | 4 bits | 91.36 (88.90) | 91.46 (88.30) | 91.53 (77.58) |
| | 3 bits | 82.35 (56.53) | 82.32 (54.17) | 82.59 (35.28) |
| Thermal noise | $\sigma = 0.01$ | 91.53 (92.82) | 83.91 (10.53) | 81.96 (10.00) |
| | $\sigma = 0.05$ | 91.37 (92.65) | 90.23 (10.28) | 70.87 (10.07) |
| | $\sigma = 0.10$ | 88.86 (91.65) | 83.91 (10.10) | 40.35 (10.65) |
| | $\sigma = 0.15$ | 87.76 (90.83) | 71.09 (10.16) | 10.00 (10.27) |
| | $\sigma = 0.20$ | 85.77 (89.85) | 63.00 (10.37) | 10.58 (10.61) |
| Neuron silence | $p = 0.1$ | 92.31 (92.93) | 91.87 (10.48) | 10.47 (10.33) |
| | $p = 0.2$ | 91.82 (92.77) | 90.93 (10.40) | 10.26 (10.18) |
| | $p = 0.3$ | 90.91 (92.72) | 89.42 (16.44) | 10.39 (10.07) |
| | $p = 0.4$ | 89.38 (92.28) | 86.31 (28.77) | 10.52 (10.24) |
| | $p = 0.5$ | 86.53 (91.87) | 83.09 (70.35) | 10.23 (10.29) |