# OpenReview forum: "Training Spiking Neural Networks with Local Tandem Learning"
_NeurIPS.cc/2022/Conference — NeurIPS 2022 Accept_

### Official Review · Reviewer_6iuj · 2022-06-20

**Rating:** 8
**Confidence:** 5
**Soundness:** 3 good
**Presentation:** 4 excellent
**Contribution:** 3 good

**Summary:**

This paper proposes a generalized learning rule called Local Tandem Learning (LTL) which the SNN mimics the feature representation of a pre-trained ANN through local loss functions. The intermediate feature representation of a pre-trained ANN is transferred to SNN through layer-wise local loss functions. This method has not only high accuracy but also fast convergence It has two versions: offline and online and both versions have high performance. In the experiments, the authors verify the high performance of this method on CIFAR-10, CIFAR-100, and ImageNet dataset. Besides, this method is hardware friendly and can be easily implemented on-chip which is robust to hardware-related noises.

**Questions:**

(1)	The comparison of accuracy between the Local Tandem Learning (LTL) method and only use the surrogate gradient method to training SNN can be added to verify that this method learns the knowledge in ANN.
(2)	More architectures such as ResNet can be used to verify the high performance of this algorithm rather than just VGG network architecture in order to better demonstrate the advantages of this approach in different network structures.


**Limitations:**

The authors put forward a generalized learning rule, termed Local Tandem Learning (LTL) which the SNN mimics the feature representation of a pre-trained ANN through local loss functions. It uses the intermediate feature represents of the pre-trained ANN to supervise the training of SNNs. This method considers the problem that the SNN is difficult to train because of the discrete spikes. It improves the accuracy and computational complexity of SNN. The research of this article is very useful and valuable.

**Strengths And Weaknesses:**

Strengths:
(1)	The method proposes a local loss function trains the SNN by learning the intermediate feature representation of the middle layers of the pre-trained ANN. It combines the advantages of ANN-to-SNN conversion and gradient-based training methods. This work is very valuable.
(2)	The authors have done sufficient experiments. This method reduces the epochs required for training to convergence. It can achieve rapid network convergence within five training epochs on the CIFAR-10 dataset.
(3)	The accuracy of SNN trained by this method gap to the teacher ANN is almost zero. Compared with other SOTA learning algorithms, the proposed methods also can achieve comparable performance.
Weaknesses:
(1)	Background knowledge of the Teacher-Student (T-S) learning approach is not introduced and some introductions about this aspect can be added.

---

> ### Author Response · Authors · 2022-08-02
> **Response to Reviewer 6iuj**
>
> Thank you very much for your thoughtful and constructive comments. We are delighted that you find our work to be valuable and the experiments are sufficient. We would like to address your specific concerns as follows:
>
> >  Background knowledge of the Teacher-Student (T-S) learning approach is not introduced and some introductions about this aspect can be added.
>
> Thank you for bringing this up. We will bridge this information gap by including the following paragraph into the Introduction section.
>
> *As a generalized knowledge transfer approach, teacher-student (T-S) learning leverages the expert teacher model to guide the training of the student model. Hinton et al. [1] first proposed the idea, known as knowledge distillation, to transfer the knowledge from a  model ensemble or a large, regularized model to a small, distilled model so as to achieve effective model compression. Romero et al. [2] further proposed FitNets, which enhanced the knowledge distillation framework by transferring the fine-grained, intermediate feature maps of the teacher to the student model. Later, Huang et al. [3] proposed to enforce the student network to mimic the feature distribution of the teacher network. The T-S learning approach was originally introduced for model compression [1,2], and subsequent work extended it for other task domains, including domain adaptation [4], ensemble learning [5]. For a more complete review on the T-S Learning, we refer the readers to [6]. In this work, we apply this approach to heterogeneous networks ( i.e., ANN and SNN) for the first time, and we show information can be transferred effectively and efficiently from teacher ANN to the student SNN with our proposed LTL learning rule.*
>
> [1] Hinton, Geoffrey, Oriol Vinyals, and Jeff Dean. "Distilling the knowledge in a neural network." arXiv preprint arXiv:1503.02531 2.7 (2015). \
> [2] Romero, Adriana, et al. "Fitnets: Hints for thin deep nets." arXiv preprint arXiv:1412.6550 (2014). \
> [3] Huang, Zehao, and Naiyan Wang. "Like what you like: Knowledge distill via neuron selectivity transfer." arXiv preprint arXiv:1707.01219 (2017). \
> [4] Lu, Jie, et al. "Transfer learning using computational intelligence: A survey." Knowledge-Based Systems 80 (2015): 14-23. \
> [5] Kang, Jaeyong, and Jeonghwan Gwak. "Ensemble learning of lightweight deep learning models using knowledge distillation for image classification." Mathematics 8.10 (2020): 1652. \
> [6] Gou, Jianping, et al. "Knowledge distillation: A survey." International Journal of Computer Vision 129.6 (2021): 1789-1819.
>
> > The comparison of accuracy between the Local Tandem Learning (LTL) method and only use the surrogate gradient method to training SNN can be added to verify that this method learns the knowledge in ANN.
>
> Thank you for the suggestion. We have included a more recent direct training method, TET[1], for a more concrete comparison. The experiments results can be found on the reponse to the Reviewer XCfF provided on top. In summary,  our results are comparable to the TET method on the CIFAR-10 dataset and CIFAR-100 dataset, while requiring a slightly longer inference time window due to the underlying rate-based representation. These results align with our previous comparison to the direct training method: Diet-SNN.
>
> [1] Deng, S., Li, Y., Zhang, S., & Gu, S. (2022). Temporal Efficient Training of Spiking Neural Network via Gradient Re-weighting. ICLR22.
>
> > More architectures such as ResNet can be used to verify the high performance of this algorithm rather than just VGG network architecture in order to better demonstrate the advantages of this approach in different network
>
> Thank you for the useful suggestion. As summarised in the table below, we have performed additional experiments on the ResNet architectures to evaluate the generalizability of our work. It is clear that the LTL learning rule can achieve superior classification accuracies on the ResNet architecture. Due to time constraints, we only manage to obtain the results for the offline setting and we will include the online results in the camera-ready version.
>
> | Dataset | Method | Model | ANN Acc. (%) | SNN (IF) Acc. (%)  | SNN (LIF) Acc. (%)  | T |
> | --------- | --------- | --------- | --------- | --------- | --------- | --------- |
> | CIFAR-10          | LTL (offline) | ResNet-20 | 95.36 | 94.82 | 94.76 | 16 |
> | CIFAR-10          | LTL (offline) | ResNet-20 | 95.36 | 95.28 | 95.25 | 32 |
> | CIFAR-100        | LTL (offline) | ResNet-20 | 76.36 | 75.22 | 75.62 | 16 |
> | CIFAR-100        | LTL (offline) | ResNet-20 | 76.36 | 76.08 | 76.12 | 32 |
> | Tiny-ImageNet  | LTL (offline) | ResNet-20 | 56.68 | 56.28 | 56.24 | 16 |
> | Tiny-ImageNet  | LTL (offline) | ResNet-20 | 56.68 | 56.48 | 57.42 | 32 |

---

> > ### Comment · Reviewer_6iuj · 2022-08-07
> > **Improve my score**
> >
> > I am very happy to see the authors have made great improvements in experiments by including on-chip learning implementation and substantial experimental comparison with other SOTA methods. This work will be of sufficient significance to advance the field of neuromorphic computing.

---

### Official Review · Reviewer_XCfF · 2022-06-27

**Rating:** 8
**Confidence:** 4
**Soundness:** 4 excellent
**Presentation:** 4 excellent
**Contribution:** 3 good

**Summary:**

Summary:
This paper proposes a spiking neural network (SNN) learning rule, namely Local Tandem Learning (LTL). It combines the teacher-student learning framework and a novel layer-wise local learning method for ANN-to-SNN conversions. Experiments demonstrate near lossless knowledge transfer can be achieved on Cifar-10, Cifar-100 and Tiny-ImageNet datasets with fast convergence and low computational complexity. Notably, authors also show that the LTL rule is hardware friendly. It can address the hardware-related noises through on-chip learning, which remains a grand challenge for deploying SNNs on the ultra-low-power mixed-signal neuromorphic computing chips.

Overall, this paper is well-written and easy to understand. In terms of technical quality, the proposed local tandem learning rule is novel, significant, and well-motivated. It provides an appreciable advance in the field of neuromorphic computing.





**Questions:**

1. Please summarize the key differences from previous ANN-to-SNN work[1]
2. Please fix the typo of one of the works in Table 1: OCFS

Ref.
Wu J, Chua Y, Zhang M, et al. A tandem learning rule for effective training and rapid inference of deep spiking neural networks[J]. IEEE Transactions on Neural Networks and Learning Systems, 2021.

**Ethics Review Area:**

["I don’t know"]

**Strengths And Weaknesses:**

Pros:
The topic of this paper is of sufficient interests to the neuromorphic computing community. As clearly summarized in the paper, the LTL learning rule has several attractive feats:
1.	The proposed Teacher-Student learning framework is a generalized knowledge transfer approach. It can be applied for different neuron models, which is out of the reach of the existing ANN-to-SNN conversion methods.
2.	The proposed local learning approach is a valuable exploration beyond the mainstream e2e SNN training algorithms. By decoupling the learning process for each layer, the proposed method reduces the required spatial-temporal credit assignment to only the temporal domain, which greatly improve the network convergence speed and can also alleviate the compounding gradient approximation errors of the surrogate gradient method.
3.	This paper also addresses an under-explored direction of algorithm-hardware co-development. This paper shows the proposed LTL rule can be easily implemented on-chip using only locally available information.

Cons:
1.	For experimental results in Table 1, the comparisons are mainly focus on the ANN-to-SNN conversion methods, while authors only compared with one direct training method. I suggest authors to include other more recent direct training algorithms to show the superiority.
2.	For online LTL learning, as the first few time-steps are generally unstable, with inaccurate firing rate estimation, hence result in noisy training. I am wondering how this affects the learning performance.
3.	In the hardware-related noise experiments, authors using the Gaussian noise to simulate the parameter mismatch and thermal noise. However, in the real hardware implementation, these noises may not follow the Gaussian distribution.  I am curious why authors use the Gaussian Noise in this experiment.

---

> ### Author Response · Authors · 2022-08-02
> **Response to Reviewer XCfF**
>
> Thank you very much for your thoughtful and constructive comments. We are encouraged that you find our paper novel, significant, and of sufficient interest to the neuromorphic computing community. We would like to address your specific concerns as follows:
>
> >  For experimental results in Table 1, the comparisons are mainly focus on the ANN-to-SNN conversion methods, while authors only compared with one direct training method. I suggest authors to include other more recent direct training algorithms to show the superiority.
>
> Thank you for pointing this out. Following your suggestion, we have included a more recent direct training method, TET[1], in the comparison. To ensure a fair comparison, we reproduce the results using their published codes while only modifying the network structures to be the same as ours.  As summarised in the Table below, our results are comparable to the TET method on the CIFAR-10 dataset and CIFAR-100 dataset, while requiring a slightly longer inference time window due to the underlying rate-based representation. These results align with our previous comparison to the direct training method, Diet-SNN.
>
> | Dataset | Method | Model | Acc. (%) | Time Window |
> | --------- | --------- | --------- | --------- | --------- |
> | CIFAR-10 | TET | VGG11 | 92.98 | 6 |
> | CIFAR-10 | LTL (offline) | VGG11 | 93.20 | 16 |
> | CIFAR-10 | TET | VGG16 | 93.49 | 6 |
> | CIFAR-10 | LTL (offline)  | VGG16 | 93.23 | 16 |
> | CIFAR-100 | TET | VGG16 | 72.45 | 6 |
> | CIFAR-100 | LTL (offline) | VGG16 | 74.19 | 16 |
>
> [1] Deng, S., Li, Y., Zhang, S., & Gu, S. (2022). Temporal Efficient Training of Spiking Neural Network via Gradient Re-weighting. ICLR22.
>
> > For online LTL learning, as the first few time-steps are generally unstable, with inaccurate firing rate estimation, hence result in noisy training. I am wondering how this affects the learning performance.
>
> We agree with you that the first few time-steps are generally unstable due to relatively inaccurate firing rate approximation. However, this issue can be easily overcome by treating the first few steps as warm-up steps and only performing parameter updates in the later time steps. To shed light on this, we perform an ablation study on the CIFAR-10 dataset with VGG16 by varying the number of warm-up time steps **Twarm**. It is clear that increasing the warm-up period will lead to better gradient approximation as evidenced by the improved test accuracy. In practice, we set **Twarm** to 14 for all our experiments under the online setting, which also reduces the training cost.
>
> | Twarm (online) |  SNN (LIF) | SNN (IF)  |
> | --------- | --------- | --------- |
> | 2 | 91.07 | 90.38 |
> | 4 | 92.27 | 92.20 |
> | 6 | 92.54 | 92.37 |
> | 8 | 92.71 | 92.26 |
> | 10 | 92.73 | 92.45 |
> | 12 | 92.74 | 92.15 |
> | 14 | 93.00 | 92.60 |
>
> > In the hardware-related noise experiments, authors using the Gaussian noise to simulate the parameter mismatch and thermal noise. However, in the real hardware implementation, these noises may not follow the Gaussian distribution. I am curious why authors use the Gaussian Noise in this experiment.
>
> As mentioned in Section 3.5,  we use the noise models introduced by Julian et al. [2]. According to their paper, these noise models are constructed using real data measured on the DYNA-SE, which is a mixed-signal neuromorphic processor. Therefore, it can faithfully reflect the actual hardware noises that one can expect in the real scenario.
>
> [2] Julian Büchel, Dmitrii Zendrikov, Sergio Solinas, Giacomo Indiveri, and Dylan R Muir. Supervised training of spiking neural networks for robust deployment on mixed-signal neuromorphic processors. Scientific reports, 11(1):1–12, 2021.
>
> > Please summarize the key differences from previous ANN-to-SNN work [1]
>
> Thank you for pointing out this excellent reference. Our proposed LTL learning method differs from this one in three aspects. Frist, their method needs to establish a mapping relationship for each synapse so as to achieve weight sharing between ANN and SNN. In contrast, our method only establishes the equivalence at the neuron level, rather than at each synapse, hence it has a higher level of freedom and is more generalizable to different underlying neuron models. Second, their method uses ANN activations to approximate the spike count of the SNN, and perform E2E backpropagation on the ANN. In contrast, our method eliminates the E2E backpropagation by using the layer-wise supervision signal to guide the learning of SNN. Third, their method learns from the spike train level representation and hence, ignores the temporal dynamic in SNN. Our method adopts the BPTT for each layer to address the temporal credit assignment issue, hence considering accurate temporal dynamics during training.
>
> > Please fix the typo of one of the works in Table 1: OCFS
>
> Thank you very much for pointing out this typo. We have corrected it accordingly in the manuscript.

---

> ### Comment · Reviewer_XCfF · 2022-08-07
> **Acknowledgment of rebuttal**
>
> I thank the reviewers for the rebuttal. I am satisfied with the authors‘ reply so I maintain the same rating for acceptance.

---

### Official Review · Reviewer_EHpf · 2022-07-02

**Rating:** 6
**Confidence:** 4
**Soundness:** 3 good
**Presentation:** 3 good
**Contribution:** 3 good

**Summary:**

This paper proposes a Local Tandem Learning (LTL) method to train spiking neural networks by distilling knowledge from a pre-trained ANN in a layer-wise way. Two versions of LTL, offline learning and online learning, are introduced and the on-chip implementation is discussed. Experiments on CIFAR-10, CIFAR-100, and Tiny ImageNet show competitive performance and robustness to hardware-related noises.

**Questions:**

See Weakness above.

**Limitations:**

There is no potential negative societal impact.

**Strengths And Weaknesses:**

Strengths:
1. The proposed LTL method considers both offline and online learning, and the on-chip implementation is discussed. It is novel to transfer knowledge of ANNs by distilling intermediate feature representations instead of reusing network weights.
2. Experiments from many perspectives are conducted, including classification accuracy, synaptic operations, convergence of learning, and robustness to hardware-related noises.

Weakness:
1. This paper emphasizes the on-chip implementation throughout the paper, but no real on-chip implementation is demonstrated, and more importantly, it is not explained clearly how conceptually the learning from a pre-trained ANN (rather than direct SNN training) can be implemented on chips. Figure 1(a) shows that the inference of both SNN and ANN is required in order to calculate the loss for each layer. But how can the inference of ANNs be implemented on neuromorphic chips without conversion to SNNs? Or is the inference of ANNs realized on common hardware and the signal is passed to chips? If so, is it only partly on-chip learning, and what is the advantage over off-chip learning since it still requires much off-chip computation (the inference of ANNs)? If the motivation is to address the real device mismatch problem, it is better to show some preliminary real on-chip results, otherwise the simulation of noises can also be leveraged by off-chip methods for finetuning. Or some comparison with finetuning by other off-chip methods in this simulation scenario is ok. There is some related work to address the real device mismatch problem [1]. They combine the on-chip and off-chip computation to correct real device mismatch by surrogate gradient learning. What is the comparison of the proposed method to their method? There should be some discussion and clarification to the above questions.

2. The state-of-the-art ANN-to-SNN conversion method outperforms the results (accuracy and latency) in this paper [3]. Since LTL also requires knowledge from pre-trained ANNs and involves more complex training procedure than conversion methods (conversion methods only need to train ANNs, LTL should first train ANNs and then train SNNs with knowledge from the pretrained ANNs), what is the advantage of LTL if ANN-to-SNN can achieve better results?

3. Experiments do not include the large-scale ImageNet dataset. Typical ANN-to-SNN conversion methods include results on large-scale datasets [2,3] and recently even direct training methods [4,5,6,7] have demonstrated competitive performance on ImageNet. Since this paper also transfer knowledge from ANNs, it would be better to include ImageNet results to show the scalability of the method.

4. Is the proposed method applicable to dynamic DVS datasets, such as CIFAR10-DVS? SNNs can be more powerful to process DVS inputs and have demonstrated strong results [4,5,6,7]. Since the proposed method relies on ANNs, is it limited to static inputs? How is the flexibility compared with direct training methods?

[1] Cramer B, Billaudelle S, Kanya S, et al. Surrogate gradients for analog neuromorphic computing. Proceedings of the National Academy of Sciences, 2022, 119(4): e2109194119.

[2] Li Y, Deng S, Dong X, et al. A free lunch from ANN: Towards efficient, accurate spiking neural networks calibration. International Conference on Machine Learning, 2021.

[3] Bu T, Fang W, Ding J, et al. Optimal ANN-SNN Conversion for High-accuracy and Ultra-low-latency Spiking Neural Networks. International Conference on Learning Representations, 2022.

[4] Zheng H, Wu Y, Deng L, et al. Going deeper with directly-trained larger spiking neural networks. Proceedings of the AAAI Conference on Artificial Intelligence, 2021.

[5] Li Y, Guo Y, Zhang S, et al. Differentiable spike: Rethinking gradient-descent for training spiking neural networks. Advances in Neural Information Processing Systems, 2021.

[6] Meng Q, Xiao M, Yan S, et al. Training High-Performance Low-Latency Spiking Neural Networks by Differentiation on Spike Representation. Proceedings of the IEEE/CVF Conference on Computer Vision and Pattern Recognition, 2022.

[7] Deng S, Li Y, Zhang S, et al. Temporal Efficient Training of Spiking Neural Network via Gradient Re-weighting. International Conference on Learning Representations, 2022.

---

> ### Author Response · Authors · 2022-08-02
> **Response to Reviewer EHpf**
>
> Thank you very much for the insightful and detailed feedback. We are encouraged that you find our method novel and our experiments comprehensive. We would like to address your specific concerns as follows:
>
> > This paper emphasizes the on-chip implementation throughout the paper, but no real on-chip implementation is demonstrated, and more importantly, it is not explained clearly how conceptually the learning from a pre-trained ANN (rather than direct SNN training) can be implemented on chips.
>
> Thank you for bringing this up and we fully agree with you.  To facilitate hardware implementation of the proposed LTL learning rule, we have revised Figure 1 and provided more details on the on-chip implementation. As shown in Figure 1, the on-chip training can be performed efficiently by simultaneously extracting the layerwise targets from ANNs running on the host computer for data batch i+1 and performing on-chip SNN training for data batch i. This is similar to the conventional ANN training on the GPUs, where the data preprocessing of the next data batch is performed on the CPU while the current data batch is used for ANN training on the GPU. The only difference is that the input data is preprocessed by the pre-trained ANN, in our case, to extract the targets for intermediate layers. Given the inference of ANN can be performed in parallel on the host computer, the overall training time is bottlenecked by the neuromorphic chip that operates in sequential mode, where only one sample is been processed at one time.
>
> > There is some related work to address the real device mismatch problem [1]. They combine the on-chip and off-chip computation to correct real device mismatch by surrogate gradient learning. What is the comparison of the proposed method to their method?
>
> Thank you for the excellent reference. We would like to highlight that the proposed LTL learning rule is more hardware friendly compared to the hardware-in-the-loop training approach introduced in [1]. As explained in this paper, the hardware-in-the-loop training requires two-way information communication, including reading and communicating intermediate neuronal states from the NC chip to the host computer to perform off-chip training and synchronizing the updated weights to the NC chips. Given the sequential nature of these two processes and the high implementation cost for reading and communicating neuron states (e.g., requiring to implement costly analog-to-digital converters for analog spiking neurons), our method is expected to have much lower hardware and time complexity.
>
> Moreover, the proposed layerwise training approach has better noise robustness and scalability over this end-to-end training approach. Specifically, we compare the online LTL rule to this in-the-loop training rule in addressing the **Device mismatch** introduced in Section 3.5. In particular, we study their robustness to different levels of noise and scalability to different network depths. As the results summarised in the Table below, the LTL rule is highly robust to different levels of noise and can scale up to deeper network structures. In contrast, the performance degrades for the in-the-loop learning method. This can be explained by the fact that the gradients estimated at each layer tend to be noisy, and the errors accumulated across layers during training. Whereas, our layerwise training approach can effectively overcome this problem. Due to time constraints, we are unable to provide analysis for other noise types studied in Section 3.5, but we will include them in the final version.
>
> | Model | SNN Pre-trained (software) | Noise level | Non fine-tuned |  LTL fine-tuned | E2E fine-tuned |
> | ---------| --------- | --------- | --------- |  --------- | --------- |
> | VGG9 | 99.45% | std_p = 0.5 | 99.00% |  99.42% | 99.02% |
> |  |  | std_p = 1.0 | 95.99% |  99.36% | 97.31% |
> |  |  | std_p = 1.5 | 70.51% |  99.34%| 78.40% |
> | VGG11 | 99.55% | std_p = 0.5 | 98.29% |  99.69% | 99.40% |
> |  |  | std_p = 1.0 | 87.63% |  99.60% | 86.52% |
> |  |  | std_p = 1.5 | 39.21% |  99.52% | 10.42% |
>
> In summary, we believe the proposed LTL learning rule is both effective and efficient for addressing the device non-idealities of mixed-signal NC chips. Given the limited access to the functioning mixed-signal NC chips, we wish to broadcast the proposed LTL  learning rule to the broader neuromorphic computing community, so as to allow interested hardware groups to implement and test it out on the mixed-signal NC chip.
>
> [1] Cramer B, Billaudelle S, Kanya S, et al. Surrogate gradients for analog neuromorphic computing. Proceedings of the National Academy of Sciences, 2022, 119(4): e2109194119.

---

> > ### Author Response · Authors · 2022-08-02
> > **Response to Reviewer EHpf**
> >
> > > What is the advantage of LTL if ANN-to-SNN can achieve better results?
> >
> > The ANN-to-SNN conversion methods establish a firing-rate approximation only for the non-leaky IF neurons, and the performance typically degrades for other neuron models with richer neuronal dynamics. This limits their applications to the existing neuromorphic computing chips, especially for those using analog spiking neurons.  In contrast, our method is more generalizable and we have demonstrated competitive results for both IF and LIF neuron models.
> >
> > > Experiments do not include the large-scale ImageNet dataset.
> >
> > Thank you for pointing this out and we agree with you that it would be better to include ImageNet results to facilitate comparison. However, the computational cost of conducting ImageNet experiments is beyond the reach of our lab. As informed by the author of ref [1], training a state-of-the-art SNN on the ImageNet dataset requires 112 GPU days (8 cards * 14 days) with a time window of only 6. Instead, we use the Tiny-ImageNet which can be considered as an effective surrogate for the full ImageNet dataset to test out the scalability of the proposed learning rule in more diversified, real-world scenarios. In order to provide a more comprehensive comparison, we have added a comparison to another **two** SOTA ANN-to-SNN conversion methods as summarised in the Table below. It is clear that our method can achieve better classification accuracies given the same time window.
> >
> > | Method | Architecture | ANN Acc. (%) |  T |  SNN Acc. (%) | $\Delta$ Acc. (%) |
> > | --------- | --------- | --------- |  --------- | --------- | --------- |
> > | Calibration [2] | VGG-16 | 60.95 |  16 |  44.39 | -16.56  |
> > | Calibration [2] | VGG-16 | 60.95 | 32 |  53.96 | -6.99  |
> > | QCFS [3] | VGG-16 | 57.82 |  16 |  44.53 | -13.29  |
> > | QCFS [3] | VGG-16 | 57.82 | 32 |  53.54 | -4.28  |
> > | Offline LTL | VGG-16 | 57.39 |  16 |  56.85 | -0.54  |
> > | Offline LTL | VGG-16 | 57.39 | 32 | 57.73 | 0.34  |
> > | Calibration [2] | ResNet-20 | 58.29 |  16 |  44.89 | -13.40 |
> > | Calibration [2] | ResNet-20 | 58.29  | 32 |  52.79 | -5.5  |
> > | QCFS [3] | ResNet-20  | 55.81 |  16 |  52.32 | -3.49  |
> > | QCFS [3] | ResNet-20  | 55.81 | 32 |  55.06| -0.75  |
> > | Offline LTL | ResNet-20  | 56.68 |  16 |  56.24 | -0.44  |
> > | Offline LTL | ResNet-20  | 56.68 |  32 | 57.42 | 0.74  |
> >
> > [1] Hu, Y., Wu, Y., Deng, L., & Li, G. (2021). Advancing Residual Learning towards Powerful Deep Spiking Neural Networks. arXiv preprint arXiv:2112.08954
> >
> > [2] Bu, T.; Fang, W.; Ding, J.; Dai, P.; Yu, Z.; and Huang, T. 2022. Optimal ANN-SNN Conversion for High-accuracy and Ultra-low-latency Spiking Neural Networks. In International Conference on Learning Representations.
> >
> > [3] Li, Y.; Deng, S.; Dong, X.; Gong, R.; and Gu, S. 2021. A free lunch from ANN: Towards efficient, accurate spiking neural networks calibration. In International Conference on Machine Learning, 6316–6325. PMLR.
> >
> >
> > > Since the proposed method relies on ANNs, is it limited to static inputs? How is the flexibility compared with direct training methods?
> >
> > You have rightly pointed out the limitation of the proposed method. As already discussed in the conclusion section, our method is not applicable for the dynamic datasets, but we would like to mention that the proposed teacher-student learning approach is a generalized information transfer method. In future work, we are interested to explore and apply the proposed teacher-student learning approach to the recurrent neural network and transformer architectures that are designed to handle temporally rich signals.

---

> ### Comment · Reviewer_EHpf · 2022-08-07
> **Thank you for the response**
>
> I thank the authors for their detailed reply. The response addresses the concerns about the concept of LTL learning on chips as well as the discussion with related work, and presents promising preliminary results to deal with the device mismatch problem in comparison to E2E fine-tuning. Some weaknesses, e.g. lack of large-scale results and limitation for dynamic inputs, are also discussed in the response and the reasons are acceptable. So I think the strengths outweigh the weaknesses and I raise my score.
>
> As for the supplemented results, I have a question about the setting of the experiments, i.e. the dataset and the noise level. It seems that no results in the paper are around 99%, and the noise level in Figure 5 is from 0.05 to 0.4 while it is much larger in this supplemented results (from 0.5 to 1.5). Is it from a different dataset and with a different setting? Hope the authors could give a clearer description and perfect the comparison (e.g. in Figure 5) in the final version.
>
> A few more comments on the connection with ANN-to-SNN and possible underlying explanation for LTL. The proposed LTL method is indeed more generalizable to ANN-to-SNN as it is not limited to IF neurons, and it is promising for neuromorphic computing. But I think the reason for the success of the LTL method may still depend on some possible underlying connections with the ANN-like closed-form transformations. This is because the single-layer neural network does not have universal approximation abilities, and there is no guarantee that such layer-wise teacher-student knowledge transfer can be learned if the layer-wise transformation is arbitrary (traditional knowledge distillation in ANNs is not layer-wise but model-wise). Actually, several works have derived the closed-form transformation between LIF neurons in order to train SNNs [1, 2], and [2] shows that the approximation transformation between LIF neurons with weighted firing rates can follow the same formulation as IF neurons with firing rates. Therefore, I guess this layer-wise transformation similarity between IF/LIF neurons and ANNs encourages layer-wise knowledge transfer and leads to successful results.
>
> [1] Wu J, Chua Y, Zhang M, et al. A tandem learning rule for effective training and rapid inference of deep spiking neural networks[J]. IEEE Transactions on Neural Networks and Learning Systems, 2021.
>
> [2] Meng Q, Xiao M, Yan S, et al. Training High-Performance Low-Latency Spiking Neural Networks by Differentiation on Spike Representation. Proceedings of the IEEE/CVF Conference on Computer Vision and Pattern Recognition, 2022.

---

> > ### Author Response · Authors · 2022-08-09
> > **Response to Reviewer EHpf**
> >
> > >As for the supplemented results, I have a question about the setting of the experiments, i.e. the dataset and the noise level.
> >
> > Thank you very much for raising the question on the experiment setting. Due to time constraints, we only manage to run experiments on the MNIST dataset, which is much simpler than the CIFAR-10 dataset that had been investigated in the original paper. Given the simplicity of the dataset, the effect of noise is not as obvious as the previous one. Therefore, we purposely increase the level of noise to provide a more convincing baseline to validate the effectiveness of the proposed LTL algorithm and compare with the E2E in-the-loop training. We are still running the experiments on the CIFAR-10 dataset, and we will provide a more concrete analysis in the final camera-ready version.
> >
> >
> > >A few more comments on the connection with ANN-to-SNN and possible underlying explanation for LTL.
> >
> > Thank you for sharing your insight and providing these excellent references. We agree with you that the success of the LTL method is because of the underlying connections with the ANN-like closed-form transformations. In fact, our formulation is also grounded on the firing rate equivalence between ANN and SNN.  For the two papers you have referred to, the equivalence is established at the spike-train level instead of for each time step, which seems to be a promising direction to explore and we will study them in our future work.

---

### Official Review · Reviewer_6gNN · 2022-07-07

**Rating:** 6
**Confidence:** 3
**Soundness:** 3 good
**Presentation:** 4 excellent
**Contribution:** 4 excellent

**Summary:**

This paper proposes a novel method of training spiking neural networks (SNNs) by matching the intermediate feature representations of SNNs with pretrained ANNs. The method is on-chip and local, allowing SNNs to be learned directly on neuromorphic hardware. Experiments on image classification benchmarks demonstrate that the proposed LTL rule performs comparably to other ANN-to-SNN conversion methods. Moreover, LTL is able to train SNNs with high accuracy even under small time windows, which is important to limit the number of synaptic operations of SNNs. LTL trained SNNs can also be trained with a small number of training iterations relative to STBP. Theoretically, STBP has lower or comparable memory and time complexity to baselines. Finally, LTL is robust to a variety of noises.

**Questions:**

How does the number of synaptic operations of LTL compare with other SNN-to-ANN conversion methods?

Does the asymptotic analysis presented in section 3.4 match empirical observations?

How does the noise robustness of LTL compare with other SNN-to-ANN conversion methods?

**Limitations:**

The authors touch on some limitations of the work in the conclusion: they do not study architectures for sequential data such as RNNs and transformers. It would also be good to note any practical difficulties that might arise when implementing LTL on neuromorphic hardware.

**Strengths And Weaknesses:**

**Originality**
The proposed method appears novel. To my knowledge, this method is the first on-chip ANN-to-SNN conversion method that uses layerwise feature-based transfer of information from an ANN to an SNN.

**Quality**
The method is not theoretically motivated. However, this is not strictly necessary if there are good empirical results.

Experiments are extensive. The authors demonstrate that LTL achieves comparable performance to many baseline methods (see Table 1). However, given that one of the main advantages of LTL is its efficiency and robustness, the experiments in section 3.2 - 3.5 could be significantly expanded to highlight this. In particular, the authors may want to consider additional SNN training baselines on top of STBP. Moreover, in section 3.4 the authors study the asymptotic time and memory complexity of LTL. It would be valuable to empirically validate these asymptotic trends. Finally, the experiments in section 3.5 are interesting, but unfortunately difficult to contextualize without a baseline method to compare with.

If the authors can adequately address the concerns on the experiments, then I think the overall quality of the paper can be dramatically increased.

**Clarity**
The paper is generally well written. The mathematical notation is clear and figures are well illustrated.

**Significance**
The paper is potentially quite significant. It appears to be a more efficient and robust method of training SNNs on-chip than previous method, while maintaining high accuracy. However, further empirical experiments are needed to better justify these claims. Given that the experiments are improved, the paper would be an important contribution to the field of SNN training, and may potentially be used on actual neuromorphic chips.

---

> ### Author Response · Authors · 2022-08-02
> **Response to Reviewer 6gNN**
>
> Thank you very much for your insightful and detailed review comments. We are encouraged that you find our paper well-written and that the problem tackled in our paper will have a significant impact to the neuromorphic computing community. We would like to address your specific concerns as follows:
>
> > Given that one of the main advantages of LTL is its efficiency and robustness, the experiments in section 3.2 - 3.5 could be significantly expanded to highlight this. In particular, the authors may want to consider additional SNN training baselines on top of STBP.
>
> We agree with you on this point. Following your suggestion, in Section 3.2, we performed additional experiments on inference time energy efficiency with a recently introduced ANN-to-SNN conversion method QCFS [1] that works for the extremely short inference time window. As summarized in the Table below, the proposed LTL rule consumes fewer SynOps than this network conversion method probably due to a larger number of neurons remaining silent in our network. This can be explained by the round-down operation that has been taken when calculating the target spike count in our method.
>
> | Model | Method | T=4 | T= 8 |  T=16 | T= 32 |
> | --------- | --------- | --------- | --------- | --------- | --------- |
> | VGG11 | QCFS | 0.25 | 0.50 | 0.99 | 1.98 |
> | VGG11 | LTL | 0.21 | 0.43 | 0.87 | 1.77 |
> | VGG16 | QCFS | 0.45 | 0.92 | 1.84 | 3.67 |
> | VGG16 | LTL | 0.27 | 0.57 | 1.17 | 2.39 |
>
> In addition, to strengthen our analysis of the network convergence speed, we added the comparison to a recently introduced direct learning rule TET [2] in Section 3.3. As shown in Figure 4 of the main text, our proposed LTL rule achieved much faster network convergence to both the TET and STBP learning rules on the CIFAR-10 dataset.
>
> > The experiments in section 3.5 are interesting, but unfortunately difficult to contextualize without a baseline method to compare with.
>
> Thank you for pointing this out. To improve on this, we have compared our online LTL learning rule to a recent hardware in-the-loop surrogate gradient learning method [3]. This method reads the actual membrane potentials and output spikes from the neuromorphic chip and sends them to the host computer that runs the surrogate gradient learning method to update the network parameters in an end-to-end fashion. The updated network parameters will be synchronized to the chip after every training iteration. In this way, this costly, off-chip learning method can address the hardware non-idealities discussed in Section 3.5.
>
> Here, we compare our method to this in-the-loop training method in addressing the **Device mismatch** introduced in Section 3.5.  In particular, we study their robustness to different levels of noise and scalability to different network depths. As the results summarised in the Table below, the LTL rule is highly robust to different levels of noise and can scale up to deeper network structures. In contrast, the performance degrades for the in-the-loop learning method. This can be explained by the fact that the gradients estimated at each layer tend to be noisy, and the errors accumulated across layers during training. Whereas, our layerwise training approach can effectively overcome this problem.
>
> | Model | SNN pre-trained | Noise level | Non-trained |  LTL trained | E2E trained |
> | ---------| --------- | --------- | --------- |  --------- | --------- |
> | VGG9 | 99.45% | std_p = 0.5 | 99.00% |  99.42% | 99.02% |
> |  |  | std_p = 1.0 | 95.99% |  99.36% | 97.31% |
> |  |  | std_p = 1.5 | 70.51% |  99.34% | 78.40% |
> | VGG11 | 99.55% | std_p = 0.5 | 98.29% |  99.69% | 99.40% |
> |  |  | std_p = 1.0 | 87.63% |  99.60% | 86.52% |
> |  |  | std_p = 1.5 | 39.21% |  99.52% | 10.42% |
>
> [1] Bu, T.; Fang, W.; Ding, J.; Dai, P.; Yu, Z.; and Huang, T. 2022. Optimal ANN-SNN Conversion for High-accuracy and Ultra-low-latency Spiking Neural Networks. In International Conference on Learning Representations.
>
> [2] Deng, S.; Li, Y.; Zhang, S.; and Gu, S. 2022. Temporal Efficient Training of Spiking Neural Network via Gradient Re-weighting. arXiv preprint arXiv:2202.11946.
>
> [3] Cramer B, Billaudelle S, Kanya S, et al. Surrogate gradients for analog neuromorphic computing. Proceedings of the National Academy of Sciences, 2022, 119(4): e2109194119.

---

> > ### Author Response · Authors · 2022-08-02
> > **Response to Reviewer 6gNN**
> >
> > > In section 3.4 the authors study the asymptotic time and memory complexity of LTL. It would be valuable to empirically validate these asymptotic trends.
> >
> > Thank you for your suggestion. To provide a more concrete analysis of the memory and time complexity, we have measured the actual memory consumption and training time using the VGG11 architecture and CIFAR-10 dataset. We present the average results over 10 training epochs in the Table below. In general, the GPU memory usage scales up near linearly with the time window size (i.e., T=16) for the STBP and Offline LTL methods, which follows our theoretical analysis. Compared to the STBP rule, the offline LTL rule requires slightly more memory space for the calculation of the layer-wise loss functions. As for per epoch training time, although both offline and online LTL rules are significantly faster than STBP rule, the ratios of speed up scale poorer than the theoretical ones. We would like to acknowledge that we adopt the primitive and unoptimized Pytorch GPU kernels in our implementation. To achieve the desired theoretical speed up, it requires customized GPU kernels that can perform the training both in parallel across time and layers and we leave this as future work.
> >
> > | Method | GPU Memory Usage (MB) | Training Time per Epoch (s) |
> > | --------- | --------- | --------- |
> > | STBP         | 7072 | 489.80 |
> > | Offline LTL | 7472 | 389.32 |
> > | Online LTL | 718   | 103.30 |
> >
> > > How does the noise robustness of LTL compare with other ANN-to-SNN conversion methods?
> >
> > Both the ANN-to-SNN and LTL methods adopt firing rate-based feature representation. Given the same time window size, they are expected to achieve comparable noise robustness. While if a longer time window is adopted for the ANN-to-SNN conversion method, as normally does, it is expected to suffer more from input noises (e.g., thermal noise and neuron silencing noises). Due to time constraints, we are unable to provide a concrete experimental analysis here, but we will include it in the final version. Nevertheless, we would like to highlight that the main purpose of the noise robustness experiments is to demonstrate that our LTL learning rule can perform on-chip, noise-aware learning, such that the accuracy drop during system deployment can be effectively addressed.
> >
> > >  It would also be good to note any practical difficulties that might arise when implementing LTL on neuromorphic hardware.
> >
> > Thank you for your suggestion. To facilitate hardware implementation of the proposed LTL learning rule, we have revised Figure 1 and provided more details on the on-chip implementation. As shown in Figure 1, the on-chip training can be performed efficiently by simultaneously extracting the layerwise targets from ANNs, running on the host computer, for data batch i+1 and performing on-chip SNN training for data batch i. This is similar to the conventional ANN training on the GPUs, where the data preprocessing of the next data batch is performed on the CPU while the current data batch is used for ANN training on the GPU.  The only difference is that the input data is preprocessed by the pre-trained ANN, in our case, to extract the targets for intermediate layers. Given the inference of ANN can be performed in parallel on the host computer, the overall training time is bottlenecked by the neuromorphic chip that operates in a sequential mode, where only one sample is been processed at a time.
> >
> > We would like to highlight that the proposed LTL learning rule is more hardware friendly compared to the state-of-the-art in-the-loop training approach [1] that requires two-way information communication, including reading and communicating intermediate neuronal states from the NC chip to the host computer to perform off-chip training and synchronizing the updated weights to the NC chips. Given the sequential nature of these two processes and the high implementation cost for reading and communicating neuron states (e.g., requiring to implement costly analog-to-digital converters for analog spiking neurons), our proposed method is expected to have much lower hardware and time complexity. Moreover, as discussed earlier, the proposed layerwise learning approach demonstrates better noise robustness and scalability. In summary, we believe the proposed LTL learning rule is both effective and efficient for addressing the device non-idealities of mixed-signal NC chips. Given the limited access to the functioning mixed-signal NC chips, we wish to broadcast this information to the broader neuromorphic computing community, so as to allow interested hardware groups to test out the proposed LTL learning rule and unleash the power of mixed-signal NC hardware.
> >
> > [1] Cramer B, Billaudelle S, Kanya S, et al. Surrogate gradients for analog neuromorphic computing. Proceedings of the National Academy of Sciences, 2022, 119(4): e2109194119.

---

> > > ### Comment · Reviewer_6gNN · 2022-08-08
> > > **Thank you for your response**
> > >
> > > I appreciate the authors' effort in running additional experiments. The additional experiments are very promising and my concerns regarding the experiments are mostly addressed. Moreover, the explanations offered by the authors regarding noise robustness and hardware implementation are convincing. Thus, I am raising my score.
> > >
> > > As noted by other reviewers, testing on actual neuromorphic hardware and scaling to larger datasets like ImageNet would further strengthen the contributions, although it is understandable if it is out of scope for the current submission. Nevertheless, I believe the strengths of the paper outweigh its weaknesses.

---

### Author Response · Authors · 2022-08-02
**Summary of Review**

We thank reviewers for their thoughtful and constructive comments. We are delighted that they found our method to be novel (**R1,R2,R3**) and useful (**R1, R3, R4**), and our presentation is clear (**R1, R2, R3, R4**). We are pleased that they recognize this paper address an important and under-explored research problem of neuromorphic algorithm-hardware codevelopment (**R1, R2, R3**) and is of sufficient interest and impact to the neuromorphic computing community (**R1, R3**). We are glad they found our evaluation to be extensive and cover many perspectives (**R2, R4**), including classification accuracy, the energy efficiency of inference, the convergence speed of learning, the computational complexity of learning, and robustness to hardware-related noises. One primary concern was insufficient discussion and experimental comparison for the on-chip learning setting. We agree on this and we have fully addressed this concern in the rebuttal. In the following, we address the specific questions and we will incorporate all the feedback into the final version.

---

### Meta-Review · Area_Chair_6Sz3 · 2022-08-22

**Recommendation:** Accept
**Confidence:** Certain

**Metareview:**

This paper proposes a novel method of training spiking neural networks (SNNs) by matching the intermediate feature representations of SNNs with pre-trained ANNs. The method is on-chip and local, allowing SNNs to be learned directly on neuromorphic hardware.

All reviewers agreed that the problem that the paper target to solve is important, and the proposed method is novel. During the discussion period, the authors successfully addressed the concerns of the reviewers. Therefore, I recommend acceptance.

**Award:**

No

---

### Decision · Program_Chairs · 2022-09-14

Accept